# TensorGRaD: Tensor Gradient Robust Decomposition for Memory-Efficient Neural Operator Training

**Sebastian Loeschcke**                                                      *sbl@di.ku.dk*
*University of Copenhagen*

**David Pitt**                                                              *dpitt@caltech.edu*
*California Institute of Technology*

**Robert Joseph George**                                                  *rgeorge@caltech.edu*
*California Institute of Technology*

**Jiawei Zhao**                                                            *jwzhao@meta.com*
*Meta FAIR*

**Cheng Luo**                                                            *chengluo@caltech.edu*
*California Institute of Technology*

**Yuandong Tian**                                                  *yuandong.tian@gmail.com*
*Meta FAIR*

**Jean Kossaifi**                                                        *jkossaifi@nvidia.com*
*NVIDIA Research*

**Anima Anandkumar**                                                      *anima@caltech.edu*
*California Institute of Technology*

**Reviewed on OpenReview:** *https: // openreview. net/ forum? id= wd1pTrQFv2*

## Abstract

Scientific problems require resolving multi-scale phenomena across different resolutions and learning solution operators in infinite-dimensional function spaces. Neural operators provide a powerful framework for this, using tensor-parameterized layers to capture complex, multi-dimensional relationships. However, scaling neural operators to high-resolution problems leads to significant computational demands, making the training of industrial-scale models prohibitive. In this work, we introduce **TensorGRaD**, a novel method that directly addresses the memory challenges associated with optimizing large tensor-structured weights. Our approach, inspired by a *robust tensor decomposition*, factorizes gradients as the sum of a low-rank tensor and a sparse one to efficiently capture information within optimizer states, including outliers. Additionally, we provide a recipe for mixed precision training of TENSORGRAD, achieving further memory savings without sacrificing accuracy. We showcase the effectiveness of TENSORGRAD for solving partial differential equations (PDEs) using Fourier Neural Operators. We provide theoretical support for TENSORGRAD demonstrating its fundamental advantage over matrix-based gradient compression methods. We empirically demonstrate strong memory savings across various PDE tasks, including the challenging turbulent Navier-Stokes case at a Reynolds number of $2 \times 10^5$. TENSORGRAD reduces total memory usage by over 50% while maintaining accuracy.

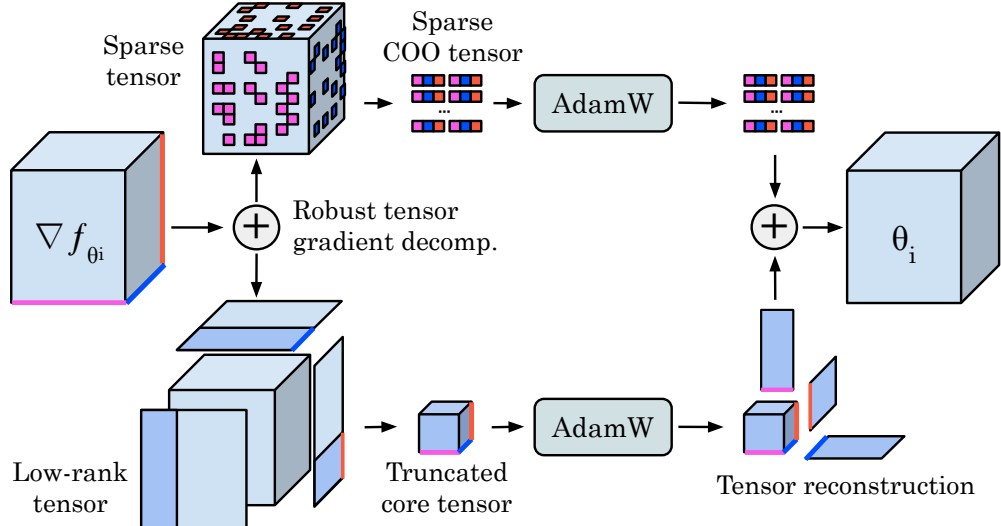

Figure 1: **Overview of TensorGRaD**. Low-rank plus sparse decomposition

# 1   Introduction

Modern deep learning has shifted towards large-scale foundation models, which have enabled unprecedented performance across diverse domains such as natural language processing, computer vision, and scientific computing (Brown et al., 2020; Kirillov et al., 2023). This represents a paradigm shift from traditional machine learning, where performance improvements are driven by scaling laws—requiring increases in data, compute, and model size (Xiao, 2025). This scaling comes at the cost of growing memory requirements. Adaptive optimizers such as Adam (Kingma and Ba, 2014), while crucial in training these large models, worsen this issue by storing additional moment tensors (e.g., first and second order moments for Adam) for each weight, which significantly increases the memory overhead (Zhao et al., 2024; Loeschcke et al., 2024a)

This memory requirement is exacerbated in the case of scientific computing, both by the size and the nature of the data and models involved. Solving scientific problems typically involves solving partial differential equations (PDEs) and resolving multi-scale phenomena on very large-scale data (Azizzadenesheli et al., 2024). This multi-dimensional data is naturally represented using tensors: multidimensional arrays that offer a natural framework for representing and manipulating complex, high-dimensional data structures (Kolda and Bader, 2009). For instance, in weather forecasting, data can span spatial grids, time steps, and atmospheric variables, leading to high-order tensor representations (Bonev et al., 2023).

**Neural operators** have been proposed as the natural framework to tackle these problems, generalizing deep learning from learning in finite-dimensional spaces to learning in function spaces (Li et al., 2023). Unlike neural networks, neural operators learn a mapping between function spaces, making them naturally suited for capturing the multi-scale structure of scientific data. To capture these multi-scale relationships, Neural Operators leverage the inherent (multi-dimensional) structure in the data, which requires maintaining high-order tensor weights and gradients to capture complex spatial, temporal, and channel interactions. As a result, unlike typical models in natural language processing or computer vision, where memory is dominated by activations, the memory overhead in neural operators is primarily driven by the tensor-structured weights and gradients. This memory requirement grows intractably with the scale of the data and has hindered the scaling of neural operators to large and complex scientific problems.

While many recent methods reduce optimizer memory in large language models, they are not directly applicable to neural operators due to the multi-dimensional structure of their weights and gradients. GaLore (Gradient Low-Rank Projection) (Zhao et al., 2024), uses a Singular Value Decomposition (SVD) to compute low-rank approximations of gradient matrices before computing optimizer states, significantly reducing memory usage during training. Extending GaLore to neural operators requires flattening gradient tensors into matrices, which disrupts their multi-dimensional structure, discarding important relationships between

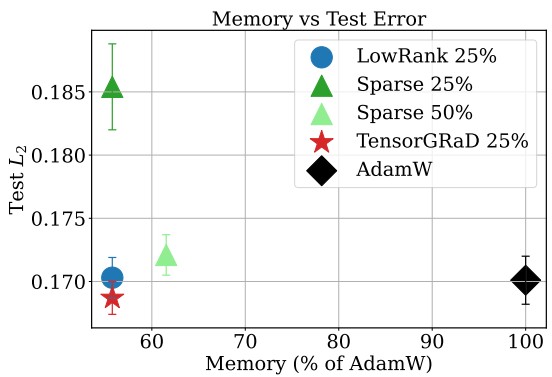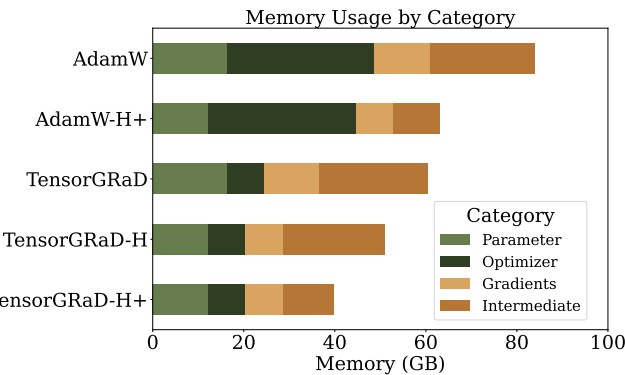

Figure 2: **Memory and performance.** Left: Comparison of low-rank, structured sparse, and TENSOR-GRAD (mixed precision) vs. Adam on Navier–Stokes $1024 \times 1024$. TENSORGRAD offers the best memory–accuracy trade-off. Right: Peak CUDA memory for FNO models with 256 channels. **H**: uses half precision for weights/gradients and full for optimizer states, and **+** includes activation checkpointing.

modes (e.g., spatial, temporal, or channel interactions). The resulting flattened matrix is not naturally low-rank and hence, standard GaLore has poor performance when low-rank structures are enforced.

This not only leads to a suboptimal low-rank approximation but can also degrade model performance, especially in scientific applications where these interactions are crucial. Alternatives to low-rank gradient projections include GRASS (Muhamed et al., 2024), which uses structured sparse projections that match low-rank methods at higher memory budgets but underperform under strict constraints. Generally, existing methods primarily focus on either low-rank or sparse representations, rarely exploring their combined application to tensor gradients, where multi-dimensional structure is critical.

**In this work**, we propose **TensorGRaD**, a novel method for efficient training of neural operators that directly addresses the memory challenges associated with tensor-structured gradients. Our approach uses an additive low-rank-plus-sparse tensor gradient compression scheme during optimization, inspired by the robust tensor decomposition viewpoint (Gu et al., 2014). Specifically, we generalize both low-rank and sparse projections to tensor gradients and combine them in an additive low-rank-plus-sparse compression scheme, inspired by robust tensor decomposition. Unlike prior approaches that apply low-rank or sparse factorization to weights or flattened gradients, TENSORGRAD operates directly on high-order tensor gradients and maintains optimizer states in decomposed form. We prove analytically that a direct extension of GaLore, relying on matricizing the gradient tensors, fails to preserve the multilinear structure required by Neural Operators. We also verify this empirically in ablation studies. We further show that under strict memory budgets, adding a small sparse residual improves over pure low-rank tensor compression by preserving residual heavy-tailed structure that low-rank projections alone do not capture well.

TENSORGRAD compresses gradients $\mathcal{G}$ using an additive decomposition $\mathcal{G} \approx \mathcal{L} + \mathcal{S}$, where $\mathcal{L}$ is a low-rank tensor approximation and $\mathcal{S}$ is a sparse residual. We emphasize that TENSORGRAD does not solve a robust tensor recovery problem at each step; rather, it uses a practical sequential compression scheme for optimizer states. We demonstrate that this low-rank-plus-sparse gradient compression remains stable under a mixed-precision strategy, running activations, weights, and gradients in half precision while maintaining optimizer states in full precision. This setup achieves substantial memory savings without compromising model accuracy. Empirically, we show that using half-precision optimizer states degrades performance, underscoring the importance of compressed full-precision optimizer states for preserving gradient information.

Implemented with AdamW, TENSORGRAD reduces memory usage by up to 75% for high-resolution neural operator learning, while maintaining comparable accuracy across several PDE benchmarks. On the challenging Navier–Stokes at $1024 \times 1024$ resolution with a Reynolds number of $2 \times 10^5$, where turbulent structures emerge across multiple scales, our mixed-precision TENSORGRAD matches the test $L_2$ loss of the full-precision Adam optimizer while reducing optimizer memory usage by up to 75% and total memory usage by more than 55%. Code is available at https://github.com/neuraloperator/tensorgrad

## 2   TensorGRaD

In this section, we first introduce the necessary background before going into detail in our method, illustrated in Fig. 1, its training, implementation, and theoretical properties.

### 2.1   Background: Tensors and Neural Operators

**Tensors** are multidimensional arrays (higher-order generalizations of vectors and matrices). An $N$th-order tensor $\mathcal{X} \in \mathbb{C}^{I_1 \times \cdots \times I_N}$ has mode sizes $\{I_n\}_{n=1}^N$. **Neural Operators** learn mappings between function spaces and are often instantiated with Fourier Neural Operators (FNOs) (Li et al., 2020) (Appendix A). FNO layers combine Fourier-mode and pointwise mixing, producing tensor-structured weights and gradients whose optimizer states dominate memory at high resolution.

### 2.2   TensorGRaD

Our method uses a low-rank-plus-sparse decomposition of the gradient tensors during training. This design is inspired by the robust tensor decomposition, in which a signal is modeled as the sum of a low-rank component and a sparse component. Robust PCA separates gross corruptions or outliers from dominant low-dimensional structure in matrices (Candès et al., 2009), and Robust Tensor Decomposition (RTD) extends this idea to higher-order tensors (Gu et al., 2014). Later work (Zhang et al., 2018) studied related low-rank-plus-sparse recovery guarantees. Unlike these recovery formulations, TENSORGRAD does not solve a robust tensor recovery objective at each step; instead, it uses this low-rank-plus-sparse perspective as a practical compression model for optimizer states.

Inspired by this modeling perspective, we compress gradients before computing and storing optimizer moments. Specifically, we use two complementary forms of structure in gradient tensors: unstructured sparsity and low-rank tensor decompositions. The sparse component preserves sharp, localized residual signals, while the low-rank component models smooth, global mode-wise structure. In practice, separating a small residual correction from the dominant low-rank subspace can also have a mild regularizing effect, analogous in spirit to the robustness motivation behind robust PCA.

**Unstructured sparse gradient tensor.**   We represent localized gradient information using a sparse COO-format tensor $\hat{\mathcal{G}}_S$ supported on a fixed set of indices $\Omega \subseteq [I_1] \times \cdots \times [I_N]$. This index set is constructed by selecting $k = \lceil \rho I \rceil$ entries from $\mathcal{G} \in \mathbb{C}^{I_1 \times \cdots \times I_N}$ according to a sparsification strategy, e.g., by inspiration from GRASS (Muhamed et al., 2024) top-$k$ magnitude, probabilistic sampling, or uniform random selection. The corresponding values are extracted to define $\hat{\mathcal{G}}_S = \mathrm{Sparse}(\mathcal{G}, \Omega)$, where $\hat{\mathcal{G}}_S$ is a $k$-nonzero sparse tensor in COO format, consisting of index–value pairs. The sparse index set $\Omega$ is recomputed only every $T$ steps and reused in between, while the sparse tensor $\hat{\mathcal{G}}_S$ is extracted from the current gradient at every step. When the sparse support changes, newly introduced indices receive zero-initialized Adam moments $(m, v)$, ensuring consistent optimizer behavior while preventing leakage of stale state across supports. Existing indices retain their accumulated moments until they are removed.

This format is compatible with standard sparse tensor operations, enabling direct addition, scaling, and indexing without reconstructing a dense tensor. Overall, this representation requires storing $k$ integer indices and $k$ complex values. It supports efficient computation in the sparse format, such as gather and scatter operations, with no dense intermediates. In practice, we keep this branch small so that it acts as a targeted residual correction; larger sparse ratios increase indexing overhead and were less effective empirically than allocating most of the budget to the low-rank branch.

**Low-rank gradient tensor decomposition.**   To compress high-dimensional gradient tensors, we use a Tucker decomposition (Tucker, 1966; Kolda and Bader, 2009), a higher-order generalization of low-rank matrix factorization (Janzamin et al., 2019). Given a tensor $\mathcal{G} \in \mathbb{C}^{I_1 \times \cdots \times I_N}$, we approximate it as $\mathcal{G} \approx [\![\mathcal{C}; U^{(1)}, \ldots, U^{(N)}]\!]$, where $\mathcal{C}$ is a core tensor of size $\mathbb{C}^{r_1 \times \cdots \times r_N}$ and $U^{(n)} \in \mathbb{C}^{I_n \times r_n}$ are orthonormal factor matrices. We compute the decomposition once and discard the core, retaining only the factor matrices. These are then reused to compress incoming gradients into a factorized representation: $\hat{\mathcal{G}}_L = \mathcal{G} \times_1 U^{(1)^\top} \cdots \times_N$

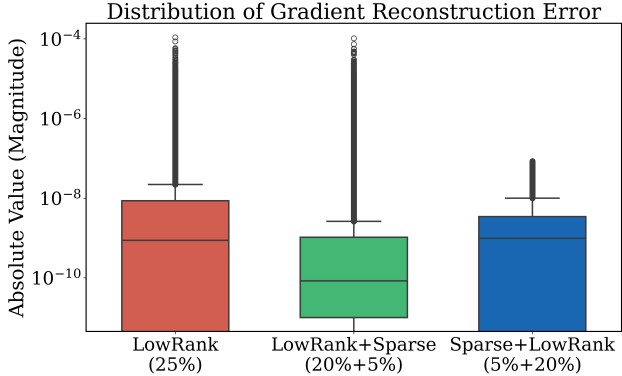 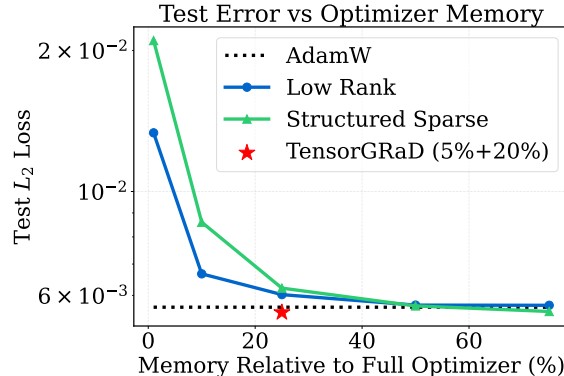

Figure 3: **Left: Gradient reconstruction error.** Box plot of gradient reconstruction error for different compression strategies on a complex FNO layer. TENSORGRAD variants reduce high-magnitude outliers. **Right: Accuracy–memory trade-off.** Performance of low-rank and structured sparse across compression ratio vs. TENSORGRAD at 25% and Adam baselines. TENSORGRAD achieves the best trade-off.

$U^{(N)^\top}$. Optimizer states are maintained directly on $\hat{\mathcal{G}}_L$, and the transformed tensor is reconstructed after the update via: $\tilde{\mathcal{G}}_L = \hat{\mathcal{G}}_L \times_1 U^{(1)} \cdots \times_N U^{(N)}$.

This decomposition reduces memory by maintaining only the factor matrices (each of size $\mathbb{C}^{I_n \times r_n}$) and the compressed optimizer state. It offers three key properties central to our method:

- **SVD generalization:** In the special case of $N = 2$, the Tucker decomposition reduces to the standard matrix SVD, linking our method naturally to GaLore.

- **Orthonormality and efficiency:** The factor matrices $U^{(n)}$ are orthonormal, allowing stable compression via mode-wise multiplication with $U^{(n)\top}$, and reconstruction using $U^{(n)}$ directly without requiring matrix inversion.

- **Structure preservation:** Tucker factorization maintains mode-wise information, avoiding the loss of semantic structure associated with tensor flattening and Kronecker approximations.

**Residual and composition.** The two components are applied sequentially. After forming the sparse or low-rank approximation $\tilde{\mathcal{G}}_1$, we compute the residual $\mathcal{R} = \mathcal{G} - \tilde{\mathcal{G}}_1$. We then use $\mathcal{R}$ to compute $\tilde{\mathcal{G}}_2$ instead of $\mathcal{G}$, i.e., after computing the residual either the low-rank decomposition or the sparse tensor is computed on $\mathcal{R}$. This composition allows each branch to focus on a different part of the gradient. Applying the sparse component first removes localized outliers and heavy-tailed residual structure, allowing the low-rank branch to model the remaining smooth, mode-wise signal more effectively. In contrast, applying low-rank compression first can leave large residual outliers for the sparse branch to correct.

**Optimizer update.** Each component is updated independently using Adam in its compressed space. First and second moment estimates $(\mathcal{M}_S, \mathcal{V}_S)$ and $(\mathcal{M}_L, \mathcal{V}_L)$ are maintained for the sparse and low-rank parts, respectively. The full update is reconstructed as:

$$\Delta \mathcal{W} = \alpha \left( \tilde{\mathcal{G}}_L + \lambda \tilde{\mathcal{G}}_S \right), \qquad \mathcal{W}_{t+1} = \mathcal{W}_t + \eta \cdot \Delta \mathcal{W}.$$

The order of the decompositions matters only during the forward pass, as the first component defines the residual for the second. For memory efficiency, the low-rank component is reconstructed first, and the sparse values are added directly into the same tensor via scatter operations. This avoids having two full tensors in memory at once. This low-rank plus sparse representation reduces memory overhead by maintaining compact optimizer states in compressed form. Unstructured sparsity captures localized outliers, while mode-wise low-rank decompositions preserve global structure. By combining them, TENSORGRAD achieves higher fidelity under strong memory constraints than either approach alone. We present the pseudocode in 1.

Fig. 3 compares the distribution of reconstruction errors for a complex gradient tensor ($64^3 \times 32$) from an FNO layer measured on a single training step of the Navier–Stokes $128 \times 128$ dataset. We evaluate three strategies:

(1) unstructured sparse followed by low-rank compression ($5\% + 20\%$), (2) low-rank followed by sparse ($20\% + 5\%$), and (3) pure low-rank compression at $25\%$. The reconstruction error is measured as the absolute difference between the original gradient $\mathcal{G}$ and its approximation $\tilde{\mathcal{G}}$, i.e., $|\mathcal{G} - \tilde{\mathcal{G}}|$. All layers exhibit similar trends. The pure low-rank method introduces more high-magnitude outliers, while the combined strategies produce tighter distributions. Applying sparse compression first yields fewer large errors but slightly higher average reconstruction error, suggesting a trade-off between error spread and overall magnitude.

### 2.3 Mixed precision training

We show that our gradient compression method remains stable when activations, weights, and gradients are computed in half precision, provided that optimizer states are maintained in full precision. This extends recent work on FNO training by Tu et al. (Tu et al., 2024), which established approximation guarantees for mixed-precision FNO training using AMP. However, their approach retains weights in full precision and casts them to half precision during computation. In contrast, we explore full half-precision training beyond AMP, including weight storage. Additionally, we find that storing optimizer states in half precision significantly degrades performance, further emphasizing projected optimizer states as an alternative. Combining mixed-precision training with TENSORGRAD allows further memory savings while maintaining accuracy.

### 2.4 Implementation

The **low-rank component** of TENSORGRAD uses the efficient Tucker decomposition from TensorLy (Kossaifi et al., 2019), implemented via Higher-Order Orthogonal Iteration. See Appendix G.1 for more details. All subsequent operations, like compressing the gradients and reconstructing the low-rank updates, are performed using PyTorch. The **unstructured sparse component** is implemented natively in PyTorch. We extract the top-$k$ or randomly sampled values based on a given sparsification strategy and store them as index–value pairs. This format supports direct operations like elementwise scaling and addition without dense reconstruction. Together with gradient compression and mixed precision training, these techniques allow TENSORGRAD to scale efficiently to large neural operators. We refer to this full setup as **TensorGRaD +**, and highlight its most memory-efficient variant in Fig. 2.

### 2.5 Theoretical Results for TensorGRaD

We analyze the structural advantage of applying low-rank projections directly to tensor modes rather than after matricizing a tensor gradient. Matrix methods such as GaLore (Zhao et al., 2024) impose a rank constraint on a single unfolding of the gradient. In contrast, TENSORGRAD projects each tensor mode separately, preserving the distinction between channel, spatial, and Fourier-mode directions. This distinction is important because low rank in one matricization does not generally imply low rank in the other tensor unfoldings. Appendix O provides the notation, Appendix Q the proof details, and Appendix R the comparison to matricized GaLore.

**Fixed-subspace model.** Our theorem isolates the tensor low-rank branch of TENSORGRAD under fixed projection subspaces. Let $\{P_k \in \mathbb{R}^{I_k \times r_k}\}_{k=1}^{d}$ be mode-wise projection matrices with orthonormal columns. Define

$$\mathsf{P}(\mathcal{G}) := \mathcal{G} \times_1 P_1^\top \times_2 \cdots \times_d P_d^\top, \qquad \mathsf{P}^\top(\mathcal{R}) := \mathcal{R} \times_1 P_1 \times_2 \cdots \times_d P_d.$$

The compressed gradient is $\mathcal{R}_t = \mathsf{P}(\mathcal{G}_t)$ and its reconstruction is $\tilde{\mathcal{G}}_t = \mathsf{P}^\top(\mathcal{R}_t)$. We analyze the projected update $\mathcal{W}_t = \mathcal{W}_{t-1} + \eta \tilde{\mathcal{G}}_{t-1}$. This fixed-subspace model corresponds to the low-rank tensor projection used in TENSORGRAD, but freezes the Tucker factors and removes the Adam/AdamW moment normalization, sparse residual branch, mixed precision, and projector refreshes used in Algorithm 1.

**Local assumptions.** Following the fixed-subspace analysis used for GaLore (Zhao et al., 2024), we assume that, locally along the iterates, the gradient has the multilinear form

$$\mathcal{G}_t = \frac{1}{N} \sum_{i=1}^{N} \left( \mathcal{A}_i(\mathcal{W}_t) - \mathcal{W}_t \times_1 B_{it}^{(1)} \times_2 \cdots \times_d B_{it}^{(d)} \right),$$

---

**Algorithm 1** TENSORGRAD: Adam with Unstructured Sparse + Tucker Low-Rank Gradient Compression

---

**Require:** Weight tensor $\mathcal{W} \in \mathbb{C}^{N_1 \times N_2 \times N_3 \times N_4}$. Step size $\eta$, scale factor $\alpha$, decay rates $\beta_1, \beta_2$, sparsity ratio $\rho \in (0,1)$, sparse scale $\lambda$, Tucker ranks $(r_1, r_2, r_3, r_4)$ (or uniform $r$), projector update gap $T$.

1: $t \leftarrow 0,\ I \leftarrow N_1 N_2 N_3 N_4,\ k \leftarrow \lceil \rho I \rceil$
2: Initialize low-rank factors $U^{(n)} \in \mathbb{C}^{N_n \times r_n}$ (e.g., random orthonormal), for $n = 1, \dots, 4$
3: Initialize low-rank Adam moments $\mathcal{M}_L, \mathcal{V}_L \in \mathbb{C}^{r_1 \times r_2 \times r_3 \times r_4} \leftarrow 0$
4: Initialize sparse index set $\Omega \leftarrow \emptyset$ and sparse Adam moments $(m_\Omega, v_\Omega) \leftarrow (0,0)$      $\triangleright\ m_\Omega, v_\Omega \in \mathbb{C}^{|\Omega|}$
5: **repeat**
6:      $\mathcal{G}_t \leftarrow -\nabla_{\mathcal{W}} \phi_t(\mathcal{W}_t)$
7:      **if** $t \bmod T = 0$ **then**
8:          $\Omega \leftarrow \mathrm{SparseIndices}(\mathcal{G}_t, k, \mathrm{strategy})$      $\triangleright$ e.g., Top-$k$, Rand-$k$, Prob-$k$
9:          $(m_\Omega, v_\Omega) \leftarrow \mathrm{ReindexMoments}(m_\Omega, v_\Omega, \Omega)$
10:         $g_\Omega \leftarrow \mathrm{Gather}(\mathcal{G}_t, \Omega)$
11:         $\mathcal{R}_L \leftarrow \mathcal{G}_t - \mathrm{Scatter}(\Omega, g_\Omega)$      $\triangleright$ dense residual
12:         $\{U^{(n)}\}_{n=1}^4 \leftarrow \mathrm{TuckerFactors}(\mathcal{R}_L, (r_1, r_2, r_3, r_4))$      $\triangleright$ HOI/HOSVD update
13:      **end if**
14:      $g_\Omega \leftarrow \mathrm{Gather}(\mathcal{G}_t, \Omega)$      $\triangleright$ unstructured sparse values (COO)
15:      $\mathcal{G}_{t,\mathrm{res}} \leftarrow \mathcal{G}_t - \mathrm{Scatter}(\Omega, g_\Omega)$      $\triangleright$ subtract sparse part (dense)
16:      $\hat{\mathcal{G}}_L \leftarrow \mathcal{G}_{t,\mathrm{res}} \times_1 U^{(1)^\top} \times_2 U^{(2)^\top} \times_3 U^{(3)^\top} \times_4 U^{(4)^\top}$      $\triangleright$ Tucker compression
17:      $(m_\Omega, v_\Omega, g_\Omega) \leftarrow \mathrm{ADAMUPDATESPARSE}(m_\Omega, v_\Omega, g_\Omega, \beta_1, \beta_2, t)$
18:      $\hat{\mathcal{G}}_L \leftarrow \mathrm{ADAMUPDATEDENSE}(\hat{\mathcal{G}}_L, \mathcal{M}_L, \mathcal{V}_L, \beta_1, \beta_2, t)$
19:      $\tilde{\mathcal{G}} \leftarrow \alpha \cdot (\hat{\mathcal{G}}_L \times_1 U^{(1)} \times_2 U^{(2)} \times_3 U^{(3)} \times_4 U^{(4)})$      $\triangleright$ reconstruct low-rank part
20:      $\tilde{\mathcal{G}} \leftarrow \mathrm{ScatterAdd}(\tilde{\mathcal{G}}, \Omega, \lambda g_\Omega)$      $\triangleright$ add sparse correction
21:      $\mathcal{W}_{t+1} \leftarrow \mathcal{W}_t + \eta \cdot \tilde{\mathcal{G}}$
22:      $t \leftarrow t + 1$
23: **until** convergence

---

where $\mathcal{A}_i$ denotes an additive term and $\{B_{it}^{(k)}\}_{k=1}^d$ are mode-wise linear operators. After projection, this model induces, for each mode $k$, a one-step recursion of the form

$$(\mathcal{R}_t)_{(k)} = (\mathcal{R}_{t-1})_{(k)} - \eta\, \mathcal{H}_{t-1}^{(k)}\big((\mathcal{R}_{t-1})_{(k)}\big) + \mathcal{E}_t^{(k)}.$$

Here $\mathcal{H}_{t-1}^{(k)}$ is the projected mode-$k$ curvature operator and $\mathcal{E}_t^{(k)}$ collects the local drift from the additive term and from changes in the mode-wise operators. We assume that the projected curvature operators are self-adjoint and positive semidefinite on the compressed Frobenius space and that, for constants $\kappa_k > 0$, $\Gamma_k < \infty$, and $\delta_k \geq 0$,

$$\left\langle X, \mathcal{H}_{t-1}^{(k)}(X) \right\rangle \geq \kappa_k \|X\|_F^2, \qquad \left\| \mathcal{H}_{t-1}^{(k)}(X) \right\|_F \leq \Gamma_k \|X\|_F, \quad \text{and} \quad \|\mathcal{E}_t^{(k)}\|_F \leq \eta\, \delta_k\, \|(\mathcal{R}_{t-1})_{(k)}\|_F.$$

The first two inequalities are projected curvature and smoothness bounds; the final inequality is a local Lipschitz/drift condition. In the explicit multilinear model above, $\delta_k$ can be taken to depend on the mode-$k$ Lipschitz constants of $\mathcal{A}_i$ and $B_{it}^{(k)}$, together with a bound on the iterates.

**Theorem 1 (Fixed-subspace contraction of tensor-mode projections)** Suppose the fixed-subspace model and local assumptions above hold for every mode $k = 1, \dots, d$, with $\delta_k < \kappa_k$. If $0 < \eta \leq \min_k \Gamma_k^{-1}$, then the compressed tensor gradients contract mode-wise:

$$\|(\mathcal{R}_t)_{(k)}\|_F \leq [1 - \eta(\kappa_k - \delta_k)]\, \|(\mathcal{R}_{t-1})_{(k)}\|_F, \qquad k = 1, \dots, d.$$

Consequently, under fixed projections, $\mathcal{R}_t \to 0$ linearly in every projected mode.

Theorem 1 concerns the tensor low-rank projection mechanism in TENSORGRAD, not convergence of the full optimizer in Algorithm 1. It shows that, under positive mode-wise curvature and controlled local drift,

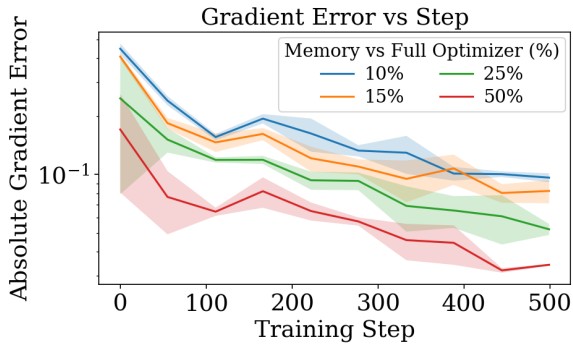 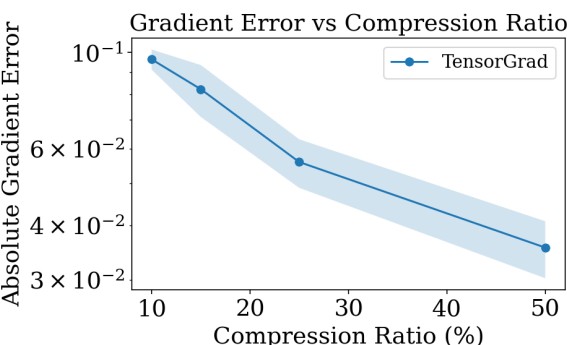

Figure 4: **Gradient reconstruction error under different compression ratios TensorGRaD. Left:** Error vs. training step for 10–50% of the full optimizer state. **Right:** Final-step error vs. memory budget (lower % = stronger compression). Both plots show mean $\pm 1$ std across three seeds.

the fixed tensor projection preserves a contractive subspace in each mode. This is precisely the structure lost by GaLore-style matricization: after flattening, the projection controls only the selected unfolding and can mix tensor modes that have distinct roles in FNO layers.

**Remark 1 (Relation to FNO layers)** In FNOs, the mode-wise operators above correspond to the local linearized effects of channel mixing, Fourier-mode mixing, and pointwise nonlinearities. Under bounded iterates, the required Lipschitz and drift conditions are standard local smoothness assumptions. For complex-valued Fourier weights, the same argument applies after replacing transposes by Hermitian adjoints.

## 3 Experimental Setup and Results

We conduct a comprehensive evaluation of TENSORGRAD on a diverse set of benchmark datasets for NOs, representing a range of PDEs with varying complexity and dimensionality.

**Datasets.** We report results on several PDE datasets: **Burgers Equation:** A one-dimensional nonlinear PDE with viscosity modeling fluid dynamics, trained on 1000 samples of Gaussian random fields at 128-point resolution. **Darcy Flow:** An elliptic PDE describing fluid flow through porous media with variable coefficients, trained on 4000 samples discretized on a $421 \times 421$ grid. **Electromagnetic Wave Propagation:** A complex-valued nonlinear Schrödinger equation modeling optical pulse propagation in waveguides with second-harmonic generation, trained on 800 samples with varying physical parameters. **Navier-Stokes:** We study the 2D Kolmogorov flow, a variant of the incompressible Navier–Stokes equations with periodic forcing (Wang et al., 2024). This dataset is particularly challenging and has a Reynolds number of $Re \approx 2 \times 10^5$, representing a highly turbulent regime. Full dataset specifications are provided in Appendix B.

**Model Architecture, Training, and Evaluation.** All models are based on the Fourier Neural Operator (FNO) architecture and trained using the Adam optimizer. Training details, including learning rates, batch sizes, and loss functions, are provided in Tab. 16. For memory profiling methodology, see Appendix J.

In evaluating TENSORGRAD, we vary the total compression ratio by adjusting the rank of the low-rank decomposition and the density of the sparse tensor to assess the trade-off between memory efficiency and performance. To further reduce memory usage, we apply **activation checkpointing** (Chen et al., 2016), which recomputes intermediate activations during backpropagation. All models are implemented in PyTorch and trained on NVIDIA A100, H100, and H200 GPUs. The main paper focuses on the most challenging Navier–Stokes at high resolution dataset and ablations of different sparse–low-rank combinations for TENSORGRAD. We present additional experiments in the appendix, including comparisons to the direct mode-unfolding extension of GaLore. We report performance using the $L_2$ test loss. Because TENSORGRAD trades memory for additional tensor decomposition work, we also report the associated runtime overhead in Figure 6 as a function of how often we update the low-rank and sparse projectors.

Table 1: **Memory and accuracy comparison on Navier–Stokes** $1024 \times 1024$ with Reynolds number $10^5$. Train and test losses are $L_2 \times 10^{-2}$ (mean $\pm$ 1 standard error over three seeds). "Mixed" uses half-precision weights and gradients with a mixed-precision forward pass. Memory is a rounded peak GPU allocation.

| Model | Rank | Memory (GB) | Precision | Train $L_2$ | Test $L_2$ |
|---|---|---|---|---|---|
| Low-Rank Only | 25% | 46 | Full | $5.37 \pm 0.08$ | $17.19 \pm 0.23$ |
| | | 29 | Mixed | $6.92 \pm 0.19$ | $17.09 \pm 0.19$ |
| GaLore (Matricized) | 25% | 46 | Full | $31.73 \pm 1.27$ | $34.56 \pm 1.43$ |
| | 50% | 49 | Full | $29.63 \pm 1.46$ | $33.11 \pm 1.21$ |
| Sparse Only | 25% | 46 | Full | $6.39 \pm 0.32$ | $18.73 \pm 0.08$ |
| | | 29 | Mixed | $7.37 \pm 0.14$ | $18.54 \pm 0.34$ |
| **TensorGRaD (ours)** | 5%+20% | 46 | Full | $5.36 \pm 0.05$ | $\mathbf{16.82} \pm 0.18$ |
| | | 29 | Mixed | $6.42 \pm 0.15$ | $16.87 \pm 0.15$ |
| Adam Baseline | 100% | 52 | Full | $3.94 \pm 0.22$ | $17.02 \pm 0.18$ |
| | | 37 | Mixed | $4.86 \pm 0.26$ | $17.01 \pm 0.19$ |

**Results on Navier–Stokes $1024 \times 1024$.** We present results on the Navier–Stokes dataset at $1024 \times 1024$ resolution with $\mathrm{Re} = 10^5$, focusing on the performance–memory trade-offs achieved by TENSORGRAD. Further results for other PDEs and detailed ablations are included in the appendix. We highlight how combining low-rank and sparse gradient compression outperforms each technique in isolation and demonstrate that TENSORGRAD is compatible with mixed-precision training.

Tab. 1 summarizes the trade-off between memory and accuracy on the challenging NS1024 dataset. At a 25% optimizer-state budget (5% unstructured sparse entries and 20% low-rank), TENSORGRAD achieves a test loss of $16.82 \times 10^{-2}$, comparable to full-precision Adam ($17.02 \times 10^{-2}$) while using substantially less optimizer memory. In comparison, low-rank-only tensor compression at the same budget yields $17.19 \times 10^{-2}$, and sparse-only compression performs worse at $18.54 \times 10^{-2}$. These results indicate that under a tight memory budget, combining a low-rank tensor approximation with a small sparse residual provides a better memory–accuracy trade-off than either component alone. By contrast, the matrix-based GaLore approach (Zhao et al., 2024), which flattens tensor gradients before projection, performs poorly on this task, with test losses of $34.56 \times 10^{-2}$ and $33.11 \times 10^{-2}$ at 25% and 50% rank, respectively. This highlights the importance of preserving tensor structure in gradient compression.

Mixed-precision training further reduces memory without harming accuracy. With TENSORGRAD, mixed-precision yields $16.87 \times 10^{-2}$ remaining comparable to full-precision Adam while reducing the total memory by 45%. We also observe that methods that match or exceed Adam in test loss often exhibit higher training loss, suggesting that compression introduces a beneficial regularization effect. This is also consistent with the sparse branch acting as a small residual correction: preserving localized directions outside the dominant low-rank subspace may help reduce mild overfitting.

Table 2: **Test $L_2$ loss $(\times 10^{-3})$ for different combinations of low-rank (LR), structured sparse (SS), and unstructured sparse (US) gradient updates**. Each cell shows top-$k$ / rand-$k$ results. Sequential forms (denoted $A \rightarrow B$) apply $A$ to the full gradient and $B$ to the residual. "Sum" applies both independently to the full gradient and sums the results.

| Method (topk/randk) | 20%+5% | 45%+5% | 5%+20% |
|---|---|---|---|
| LR $\rightarrow$ SS | 7.09 / 6.44 | 6.91 / 6.32 | 6.72 / 6.35 |
| LR $\rightarrow$ US | 6.29 / 6.20 | 6.22 / 6.19 | 6.47 / 6.26 |
| SS $\rightarrow$ LR | 6.56 / 6.26 | 6.66 / 5.96 | 6.22 / 6.21 |
| US $\rightarrow$ LR | 6.24 / 6.10 | 6.19 / 6.03 | **5.72** / 5.73 |
| LR + US (sum) | 6.18 / 6.12 | 6.17 / 6.06 | – |

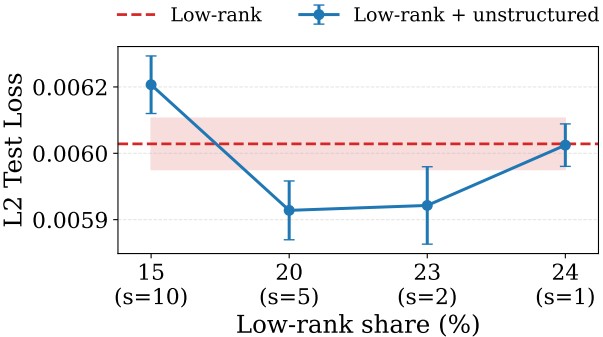

Figure 5: **Effect of sparse share at fixed 25% total budget.** Comparison of pure low-rank compression and low-rank + unstructured sparse compression at a fixed 25% optimizer-state budget. The x-axis shows the low-rank share, with the sparse share in parentheses; the y-axis reports test $L_2$ loss. Markers show the mean over three seeds and error bars one standard deviation.

**Sparse and low-rank combinations.** We evaluate different combinations of low-rank (LR) and sparse gradient compression in TENSORGRAD, varying the order, sparsity type, and selection strategy. We distinguish between structured sparsity (SS), which selects aligned slices across modes, and unstructured sparsity (US), which selects arbitrary entries. For selection strategies, we compare top-$k$ (based on magnitude) and rand-$k$ (uniform sampling). In sequential variants (denoted $A \to B$), the first component receives the full gradient and the second compresses the residual. Additive variants (denoted $A + B$) apply both directly to the gradient and sum their outputs.

The best-performing configuration is US $\to$ LR with 5% sparsity and 20% low-rank, achieving test losses of 5.72 (top-$k$) and 5.73 (rand-$k$), outperforming both compression techniques applied in isolation. In contrast, reversing the order (LR $\to$ US) leads to higher test losses (e.g., 6.29 and 6.20), indicating that removing outliers first improves the quality of the low-rank basis. Structured sparsity performs worse across all variants; for instance, LR $\to$ SS at 5% + 20% results in test errors 6.72 (top-$k$) and 6.35 (rand-$k$). This supports the role of the sparse branch as a small corrective residual: a modest sparse budget helps remove localized outliers before fitting the low-rank basis, whereas larger sparse allocations become less practical due to index storage and do not improve the overall trade-off. Despite some variants achieving similar accuracy, their practicality may differ. For example, configurations with higher unstructured sparsity require storing a large index set, increasing memory and compute overhead. In contrast, the 5% + 20% US $\to$ LR setup balances accuracy and memory.

Table 3: **Navier–Stokes ($128 \times 128$, Re $= 10^3$) precision ablation**: test $L_2 \times 10^{-3}$ under three precision schemes. **FP**: full precision. **Mixed-1**: gradients, weights, and activations (except FFT) in half precision; optimizer states in full. **Mixed-2** is identical to Mixed-1 but stores optimizer states in half precision. **LR**: low-rank; **US/SS**: unstructured/structured sparse. Lower is better; best per column is **bold**.

| Method | Full | Mixed-1 | Mixed-2 (Half Optim. states) |
|---|---|---|---|
| Adam | **5.66** | 5.62 | **6.92** |
| SGD | 9.09 | 11.12 | - |
| Adafactor | 8.19 | 8.87 | - |
| LR 50% | 5.71 | 5.70 | 7.08 |
| LR 25% | 6.02 | 5.87 | 7.21 |
| SS 50% | 5.54 | **5.56** | 7.14 |
| SS 25% | 6.22 | 6.14 | 7.78 |
| LR+US 25% (5+20) | 5.72 | 5.71 | 7.10 |

**Gradient reconstruction dynamics.** Fig. 4 shows gradient reconstruction error over training for Navier–Stokes 128 ($128 \times 128$, Re $= 10^3$) at different memory budgets. Across all budgets (10–50%, using 5% sparse

and the remainder low-rank), the error decreases over time, consistent with gradients becoming increasingly low-rank during training (Zhao et al., 2024). Stronger compression (e.g., 10%) gives higher error. Final-step error decreases monotonically with larger budgets, but this does not translate into proportional gains in task loss: Fig. 3 shows only a small $L_2$ improvement from 25% to 50% despite lower reconstruction error.

**Mixed-precision training.** We evaluate three different precision configurations on the Navier–Stokes 128 dataset ($128 \times 128$, Re $= 10^3$). In the first setting, all tensors and optimizer states are stored in full precision. The second, referred to as Mixed-1, uses half precision for weights, activations, and gradients except the Fast Fourier Transform (FFT) part (see Sec. 2.3), while keeping optimizer states in full precision. Mixed-2 is identical to Mixed-1, except that optimizer states are also stored in half precision.

Results in Tab. 3 show that in full precision, TENSORGRAD (25%) matches Adam (5.72 vs. $5.66 \times 10^{-3}$) and outperforms low-rank (25%: 6.02) and structured sparse (25%: 6.22). Structured sparsity at 50% performs best (5.54), but TENSORGRAD offers better efficiency at lower memory. When moving to Mixed-1 precision TENSORGRAD maintains strong performance, with a test loss of $5.71 \times 10^{-3}$. This matches the full-precision setting and confirms that TENSORGRAD remains stable under reduced numerical precision. Notably, some models improve slightly (e.g., Adam drops to 5.62), consistent with prior observations that mixed precision can act as a mild regularizer (Tu et al., 2024). In contrast, the Mixed-2 setup, where optimizer states are also stored in half precision, leads to significant degradation. TENSORGRAD drops to 7.10, Adam to 6.92, and low-rank (25%) to 7.21. These results underscore two key findings. First, TENSORGRAD is fully compatible with mixed-precision training, provided that optimizer states are maintained in full precision. Second, TENSORGRAD's low-rank and sparse optimizer states preserve essential gradient information more effectively than directly storing them in reduced precision. This makes TENSORGRAD a compelling approach for reducing memory without sacrificing accuracy, especially when paired with mixed precision. Our improved memory is shown in Fig. 2. In addition to Adam, we also include SGD and Adafactor (Shazeer and Stern, 2018) as memory-efficient baselines. SGD uses half the optimizer-state memory of Adam, while Adafactor uses factored second-moment statistics. On NS128, however, both perform substantially worse than TensorGRaD: in full precision, SGD and Adafactor obtain test $L_2$ losses of 9.09 and 8.19, compared to 5.72 for TensorGRaD and 5.66 for Adam; in Mixed-1, they obtain 11.12 and 8.87, compared to 5.71 for TensorGRaD and 5.62 for Adam. Implementation details and hyperparameters are provided in Appendix D.

**Effect of sparse share at fixed memory budget.** Fig. 5 isolates the contribution of the sparse branch under a fixed total optimizer-state budget of 25%, varying the split between low-rank and unstructured sparse components. A small sparse residual (around 2–5% of the full optimizer budget) improves over pure low-rank compression at matched budget, whereas allocating too much of the budget to sparsity degrades performance. This pattern is consistent with a mild regularization effect: a small sparse residual can help, whereas larger sparse shares begin to remove too much useful signal and increase indexing overhead.

**Update frequency of tensor decomposition.** Fig. 6 shows how the projector update gap $T$ affects accuracy and runtime. Performance is fairly robust over a broad range of $T$ values, consistent with prior observations for projected optimizers (Zhao et al., 2024; Muhamed et al., 2024). Very frequent updates can slightly hurt performance because repeated subspace changes introduce optimization noise and recompute imperfect decompositions more often, while very infrequent updates eventually make the projector stale. In our experiments, values around $T \in [10^2, 10^3]$ provide a good accuracy–runtime trade-off. Measured on an H100 with batch size 8 and averaged over per-epoch timing at a 25% optimizer-state budget, the slowdown at $T = 10^3$ is about 13% for pure low-rank compression and 20% for low-rank + unstructured sparse compression. The remaining overhead comes from the projection and reconstruction steps required to move gradients into and out of the compressed space.

**Additional experiments and ablations.** We show additional results in the Appendix K. Appendix K.1 evaluates the compatibility of TENSORGRAD with TFNO (Kossaifi et al., 2023), showing that optimizer-state compression remains effective even when the FNO weights are Tucker-factorized. Appendix K.2 compares TensorGRaD vs Adam on the 3D ShapeNet Car benchmark using the Geometry-Informed Neural Operator (GINO). In Appendix K.3, we ablate the effect of different update frequencies for computing factor matrices for the tensor low-rank part and for the sparse part. Appendix K.5 compares structured sparsity

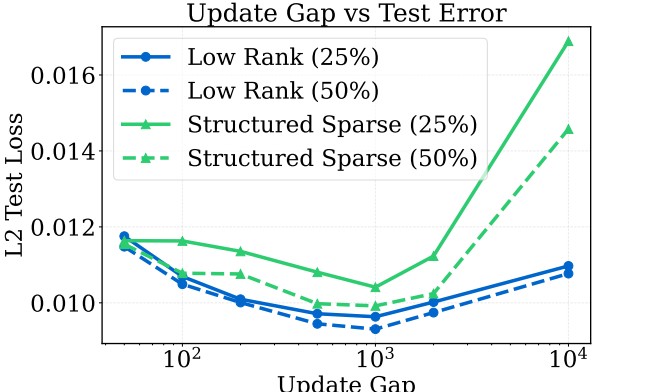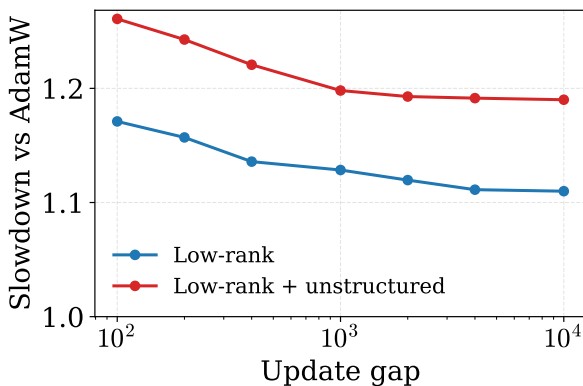

Figure 6: **Effect of projector update gap on accuracy and runtime. Left:** Test $L_2$ loss on NS128 as a function of the projector update gap $T$ for low-rank and structured sparse projections at 25% and 50% compression. **Right:** Runtime slowdown relative to AdamW as a function of $T$ for pure low-rank and low-rank + unstructured sparse compression. At $T = 10^3$, the slowdown is approximately 13% for low-rank and 20% for low-rank + unstructured sparse.

patterns using top-$k$ versus random-$k$ selection strategies. In Appendix K.4, we provide a detailed comparison between our tensor-based low-rank decomposition and a baseline method that applies a GaLore-style low-rank projection to matricized tensors. Finally, Appendix K also includes extended benchmark results across multiple datasets: Burgers, Darcy, and ElectroMagnetic.

## 4 Related Work

Our work, **TensorGRaD**, studies memory-efficient training of neural operators through tensor-aware gradient compression. While significant work has been done in related areas, the specific approach of gradient decomposition in tensors has not been explored. **Tensor Methods in Deep Learning:** Tensor decomposition has been widely used to compress deep networks (Novikov et al., 2015; Lebedev et al., 2015; Kim et al., 2016; Kossaifi et al., 2020a;b; Panagakis et al., 2021), but these methods focus on weight tensors rather than gradients during training. **Sparse gradient updates:** GRASS (Muhamed et al., 2024) introduced structured sparsity for matrices using sampling strategies like Top$-k$ magnitude sampling. In tensor settings, this approach requires unfolding tensors, disrupting the inherent mode-wise structure. Instead, we use unstructured sparsity, selecting individual tensor entries directly without unfolding them.

**Neural Operators:** Recent advancements have led to neural operators (Li et al., 2020; Kovachki et al., 2021), with FNOs showing remarkable success in scientific computing tasks. However, these methods have not explored gradient decomposition for memory efficiency. **Efficient Training Techniques:** Various approaches reduce memory footprints of large models. LoRA (Hu et al., 2022) adds a low-rank weight matrix to a frozen pre-trained matrix. FLoRA (Si et al., 2024) extends this to higher dimensions using Tucker decomposition. LoQT (Loeschcke et al., 2024a) rewrites GaLore as LoRA, optimizing only one low-rank factor in 16-bit precision and keeping the rest in 4-bit. For neural operators, MG-TFNO (Kossaifi et al., 2023) combines tensor decomposition with multi-grid approaches, while iFNO (George et al., 2024) incrementally scales FNO weight ranks during training. **Low-rank Plus Sparse:** Robust PCA (Candès et al., 2009) separates low-rank matrices from sparse noise, and this perspective has been extended to tensors by robust tensor decomposition methods (Gu et al., 2014). TensorGRaD uses a composite low-rank plus sparse approximation of the gradient, with the two components serving complementary roles during optimization. At a modeling level, this is consistent with viewing gradients as a low-rank part together with a sparse residual. Algorithmically, however, our method constructs these components sequentially through a residual-based compression procedure rather than solving a joint recovery problem. This differs from stricter tensor recovery settings that study simultaneous low-rank and sparse structure in a statistical estimation or convex recovery framework (Li et al., 2019). Hybrid low-rank-plus-sparse decompositions have been explored for model weights (Han et al., 2024) and attention matrices (Chen et al., 2021), but not for compressing gradient tensors during training. **Mixed Precision Training** utilizes lower precision formats for certain operations, reducing memory usage and potentially accelerating training on compatible hardware (Tu et al.,

2024). **Combination with existing methods** TENSORGRAD can complement many existing techniques, potentially leading to greater memory benefits by integrating with methods like FLoRA or MG-TFNO and frameworks like iFNO.

## 5 Conclusion

We presented **TensorGRaD**, a memory-efficient optimization framework for training large-scale tensor-structured models. By combining low-rank factorization with unstructured sparse gradient updates, TENSORGRAD achieves substantial memory savings without sacrificing performance. We further introduce a mixed-precision training strategy that complements our method and improves efficiency. We validate our findings on challenging PDE benchmarks and thorough ablations. Our approach enables training of large-scale neural operators on high-resolution PDE data.

**Limitations.** While TENSORGRAD delivers substantial memory savings, it also has limitations. The Tucker decomposition introduces a modest slowdown, e.g., 13–20% on NS128 dataset (see Fig. 6), even when amortized with infrequent updates. Choosing ranks, sparsity levels, and update intervals remains manual and may vary across layers or training stages. Automatically adapting these parameters remains an open problem. Finally, our evaluation focuses on PDE-driven neural operators; extending TENSORGRAD to other domains such as vision or language models is future work.

**Broader Impact.** By enabling high-resolution scientific models to train on commodity hardware, TENSORGRAD broadens access to advanced simulation tools and large-scale scientific ML. We hope it will support wider adoption of efficient neural operators in science and engineering.

## Acknowledgements

Sebastian Loeschcke is supported by the Danish Data Science Academy, which is funded by the Novo Nordisk Foundation (NNF21SA0069429) and VILLUM FONDEN (40516). Robert Joseph George is supported by a Caltech Graduate Fellowship. David Pitt is supported by the Schmidt Scholars in Software Engineering program. Anima Anandkumar is supported by the Bren Named Chair, Schmidt AI 2050 Senior fellow, and ONR (MURI grant N00014-18-12624).

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

Table 4: **Notation used in TensorGRaD.**

| Symbol | Meaning |
| --- | --- |
| $\mathcal{W}_t$ | Weight tensor at training step $t$ |
| $\mathcal{G}_t$ | Full gradient tensor at step $t$ |
| $\Omega$ | Sparse index set used for the unstructured sparse branch |
| $\hat{\mathcal{G}}_S$ | Sparse gradient representation extracted on support $\Omega$ |
| $g_\Omega$ | Values of the gradient gathered at indices $\Omega$ |
| $\mathcal{R}$ | Residual tensor after removing the first component |
| $\mathcal{R}_L$ | Residual tensor used to compute/update Tucker factors |
| $U^{(n)}$ | Tucker factor matrix for mode $n$ |
| $\hat{\mathcal{G}}_L$ | Compressed low-rank gradient in Tucker coordinates |
| $\tilde{\mathcal{G}}_L$ | Reconstructed low-rank gradient update |
| $\tilde{\mathcal{G}}_S$ | Reconstructed sparse correction |
| $\tilde{\mathcal{G}}$ | Final reconstructed gradient update applied to the weights |
| $\mathcal{M}_L, \mathcal{V}_L$ | First and second moment tensors for the low-rank branch |
| $m_\Omega, v_\Omega$ | First and second moment states for the sparse branch |
| $\rho$ | Sparse ratio, determining $k = \lceil \rho I \rceil$ retained entries |
| $k$ | Number of retained sparse entries |
| $(r_1, \ldots, r_N)$ | Tucker multilinear ranks |
| $T$ | Projector update gap / update interval |
| $\alpha$ | Scale factor applied to the reconstructed update |
| $\lambda$ | Relative scaling of the sparse correction |
| $\eta$ | Optimizer step size / learning rate |
| P | Fixed tensor projection operator used in the theory section |
| $\mathcal{A}_i$ | Additive term in the theoretical multilinear gradient model |
| $B_{it}^{(k)}$ | Mode-$k$ linear operator in the theoretical gradient model |
| $\kappa_t^{(k)}$ | Projected mode-$k$ curvature quantity in the theory |

# Appendix

# A  Neural Operators

**Neural Operators** $\mathcal{G}_\theta : \mathcal{A} \times \theta \to \mathcal{U}$ combine linear integral operators $\mathcal{K}$ with pointwise non-linear activations $\sigma$ to approximate non-linear operators, mapping initial conditions $a \in \mathcal{A}$ to solutions $u \in \mathcal{U}$. Their operation is defined as $\mathcal{G}_\theta := \mathcal{Q} \circ (W_L + \mathcal{K}_L) \circ \cdots \circ \sigma(W_1 + \mathcal{K}_1) \circ \mathcal{P}$, where $\mathcal{P}$ and $\mathcal{Q}$ are pointwise neural networks for encoding and decoding, $W_l$ are linear operators, $\mathcal{K}_l$ are integral kernel operators, and $\sigma$ are activation functions.

The **Fourier Neural Operator (FNO)** proposes a specific convolution operator for $\mathcal{K}$, defined as $(\mathcal{K}v_l)(x) = \mathcal{F}^{-1}(R \cdot T_K \mathcal{F} v_l)(x)$, where $\mathcal{F}$ and $\mathcal{F}^{-1}$ are the Fourier transform and its inverse, $R$ is a learnable transformation, and $T_K$ truncates to the lowest $K$ Fourier modes. This formulation allows FNO to be discretization-invariant, producing high-quality solutions for query points not in the training grid and enabling transfer between different grid resolutions and discretizations.

# B  Dataset

## B.1  Navier-Stokes Datasets

**Navier-Stokes 1024:**  We use the 2D Kolmogorov flow of Wang et al. (Wang et al., 2024), a periodically forced, incompressible variant of the Navier–Stokes equations. The velocity field $\mathbf{u}(x, y, t) \in \mathbb{R}^2$ evolves on a

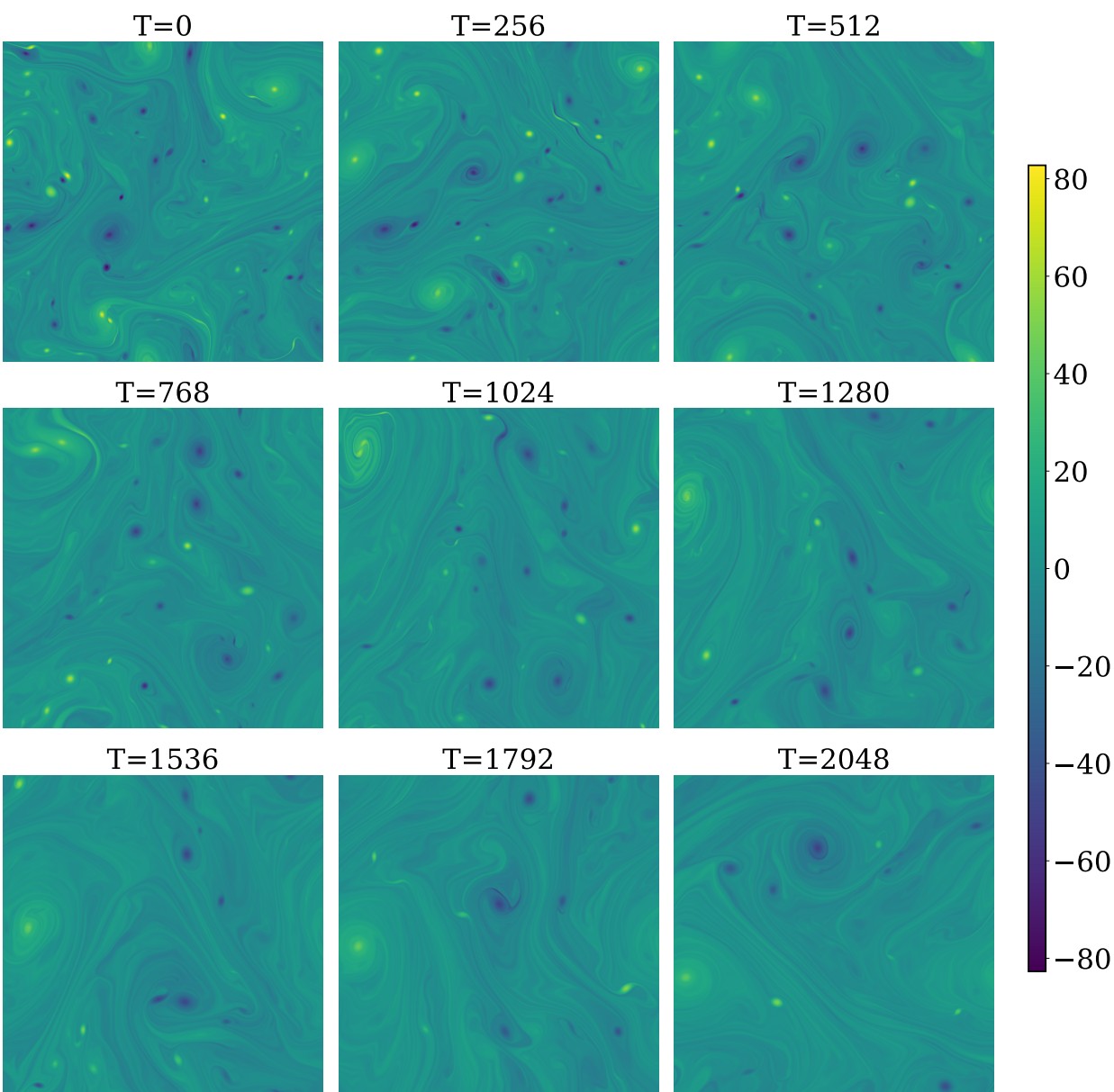

Figure 7: Navier-Stokes $1024 \times 1024$ with Reynolds number at $2 \times 10^5$

periodic domain $[0, 2\pi]^2$ according to:

$$\partial_t \mathbf{u} = -(\mathbf{u} \cdot \nabla)\mathbf{u} - \nabla p + \nu \Delta \mathbf{u} + (\sin(4y), 0)^T, \quad \nabla \cdot \mathbf{u} = 0, \quad (x, y, t) \in [0, L]^2 \times \mathbb{R}_+,$$

Our analysis focuses on the vorticity form, where the vorticity $\omega = \nabla \times \mathbf{u}$ evolves as:

$$\partial_t \omega = -\mathbf{u} \cdot \nabla \omega + \nu \Delta \omega + \nabla \times (\sin(4y), 0)^T.$$

The behavior of this flow is characterized by the Reynolds number $Re = \frac{\overline{u}l}{\nu}$, where $\nu$ is the kinematic viscosity, $\overline{u}$ is the root-mean-square velocity, and $l$ is the characteristic length. Higher $Re$ values correspond to more turbulent flows. In our setup, $L = 2\pi$ and $\nu = 10^{-4}$, leading to $Re \approx 2 \times 10^5$, representing a highly turbulent regime.

**Initial Condition and Data Collection:** The initial velocity field $\mathbf{u_0}(x)$ is sampled from a Gaussian random field $\mathcal{N}(0, C)$ with covariance

$$C = 7^{3/2}(-\Delta + 49I)^{-2.5}.$$

Data is collected from $t = 30$ onward, ensuring a statistically steady state. Our dataset, sourced from (Wang et al., 2024), comprises $[7681, 1024, 1024]$ entries, from which we generate 1577 training samples and 279 test samples, with $\Delta t = 256$. Figure 7 illustrates 9 representative samples from the dataset, selected at uniform intervals of 256 steps.

**Input-Output Pair Construction:** Input-output pairs are defined by:

- $T$: The timestep difference between the input and output.
- $t_{\text{skip}}$: The number of frames skipped between samples.

For example, with indices $[0, 1, 2, \ldots, 32]$, setting $T = 16$ and $t_{\text{skip}} = 8$ creates pairs $[(0, 16), (8, 24), (16, 32)]$. Our setup uses $T = 256$ and $t_{\text{skip}} = 4$, resulting in 1577 training samples and 279 test samples.

**Navier-Stokes (NS128):** We also experiment with the two-dimensional Navier-Stokes equation in vorticity form:

$$\partial_t \omega + \nabla^\perp \phi \cdot \omega = \frac{1}{Re} \Delta \omega + f, \quad x \in \mathbb{T}^2, t \in (0, T]$$
$$-\Delta \phi = \omega, \quad \int_{\mathbb{T}^2} \phi = 0, \quad x \in \mathbb{T}^2, t \in (0, T] \tag{1}$$

with a Reynolds number $Re = 10^3$ and final time $T = 5$. The domain is discretized on a $1024 \times 1024$ grid. We generate 10,000 training samples and 2,000 test samples using a pseudo-spectral method. To test scalability, we also evaluate our approach on a subsampled resolution of $128 \times 128$. Memory profiling is performed at the full $1024 \times 1024$ resolution.

**Burgers Equation:** We consider the one-dimensional Burgers equation on the torus:

$$\partial_t u + u u_x = \nu u_{xx}, \quad x \in \mathbb{T}, t \in (0, T] \tag{2}$$

with initial condition $u_0 \in L^2(\mathbb{T}; \mathbb{C})$ and viscosity $\nu > 0$. We set $T = 1$ and $\nu = 0.01$. Input functions are sampled from a Gaussian random field, and solutions are obtained using a pseudo-spectral method. We use 1000 samples for training and 200 for testing, with 128 resolution.

**Darcy Flow:** The Darcy flow problem is defined by the elliptic PDE:

$$-\nabla \cdot (a(x)\nabla u(x)) = f(x), \quad x \in (0, 1)^2 \tag{3}$$

with boundary conditions $u(x) = 0$ for $x \in \partial(0, 1)^2$. The input $a$ is sampled from a Gaussian random field, and $f$ is fixed. We use 4000 training samples and 1000 test samples, with the domain discretized on a $421 \times 421$ grid.

**Electromagnetic Wave Propagation:**  Lastly, we present a dataset that represents complex-valued data inherently. We consider the propagation of optical pulses in a nonlinear waveguide with second-order nonlinearity ($\kappa^2$). The problem is governed by the nonlinear Schrödinger equation (NLSE) with additional terms for second-harmonic generation:

$$\frac{\partial A}{\partial z} = -\mathrm{i}\frac{\beta_2}{2}\frac{\partial^2 A}{\partial t^2} + \mathrm{i}\gamma|A|^2 A + \mathrm{i}\kappa A^* e^{\mathrm{i}\Delta k z} \tag{4}$$

where $A$ is the complex electric field envelope, i is the imaginary unit, $z$ is the propagation distance, $t$ is time, $\beta_2$ is the group velocity dispersion, $\gamma$ is the nonlinear parameter, $\kappa$ is the coupling coefficient for second-harmonic generation, and $\Delta k$ is the phase mismatch. Our dataset consists of 800 training samples and 200 testing samples. The input consists of several parameters: the poling region length ranging from 2mm to 15mm, the poling period mismatch varying from -50nm to +50nm, and the pump pulse energy spanning from a few fJ to thousands of fJ. Additionally, the input includes the complex electric field envelope of the input pulse. The output of the system is the complex electric field envelope of the resulting output pulse.

## C  Practical hyperparameter guidance

We choose TensorGRaD hyperparameters from a target optimizer-state memory budget and then refine the low-rank/sparse split using the ablations in the main text. In our experiments, a total optimizer-state budget of about 25% of Adam gives the best memory–accuracy trade-off: performance improves substantially up to this regime and then largely saturates (Fig. 3, right).

Within this budget, most of the memory should be allocated to the low-rank branch. The fixed-budget sparse-share ablation (Fig. 5) shows that adding a small unstructured sparse residual improves over pure low-rank compression at matched budget, while allocating too much of the budget to sparsity degrades performance and increases indexing overhead. In practice, we therefore recommend keeping the sparse share small, typically around 2–5% of the full Adam optimizer state, and assigning the remainder to the low-rank component. Our default setting is 20% low-rank plus 5% unstructured sparse.

For the projector update gap $T$, the update-frequency ablation (Fig. 5) shows that performance is fairly robust over a broad range. Very frequent updates increase runtime and can slightly hurt performance due to repeated subspace switching, while very infrequent updates eventually make the projector stale. In our experiments, values in the range $T \in [10^2, 10^3]$ provide a good memory–accuracy–time trade-off, and we use $T = 10^3$ as a robust default in larger-scale runs.

Overall, a simple default recipe is: start from a 25% optimizer-state budget, allocate most of it to the low-rank branch, keep the sparse branch small (around 2-5%), and choose $T$ in the stable range $10^2$–$10^3$.

We keep $\lambda$ and $\alpha$ fixed across the main experiments rather than tuning them per dataset. Here, $\lambda$ weights the sparse branch relative to the low-rank branch, while $\alpha$ scales the reconstructed compressed update. Both are set to 1.

## D  Adafactor and SGD baseline details

For SGD, we sweep learning rates $\{0.1, 0.03, 0.01, 0.003, 0.001\}$ and momentum values $\{0.9, 0.95, 0.99\}$. For Adafactor, we sweep the same learning rates and additionally tune the second-moment decay exponent $\{-0.5, -0.8, -1.0\}$ and clipping threshold $\{0.5, 1.0, 2.0\}$. We report the best-performing configuration for each baseline in the NS128 comparison.

We implement Adafactor following the PyTorch formulation (Shazeer and Stern, 2018), adapted to support complex-valued parameters. Unlike Adam, Adafactor does not maintain a first-moment accumulator, and instead uses only a factored second-moment estimate together with relative step-size scaling and update clipping.

For a parameter of shape $(d_1, \ldots, d_{k-2}, m, n)$, we apply Adafactor in a batched matrix fashion: the leading dimensions $(d_1, \ldots, d_{k-2})$ are treated as batch indices, while only the last two dimensions are factor-

ized. Concretely, we store row and column second-moment statistics of shapes $(d_1, \ldots, d_{k-2}, m, 1)$ and $(d_1, \ldots, d_{k-2}, 1, n)$, and reconstruct an approximate elementwise variance from their product. Thus, our Adafactor baseline uses matrix-style factorization on tensor slices rather than a true multi-mode tensor factorization. For complex parameters, the second moment is computed from squared magnitudes $|g|^2 = g\bar{g}$, and the stored statistics are real-valued.

Compared to Adam, which stores two dense state tensors of the same shape as the parameter, Adafactor uses less optimizer-state memory. For a parameter of shape $(d_1, \ldots, d_{k-2}, m, n)$ with $B = \prod_{i=1}^{k-2} d_i$, Adam stores $2Bmn$ state values, whereas Adafactor stores only $B(m+n)$. The optimizer-state memory ratio relative to Adam is therefore

$$\frac{B(m+n)}{2Bmn} = \frac{m+n}{2mn},$$

which reduces to $1/n$ for square $n \times n$ slices.

## E  FNO Memory Usage

Fig. 8 illustrates the memory usage patterns in Fourier Neural Operators (FNOs) as the number of modes increases. This analysis provides crucial insights into the scalability challenges faced when training large FNO models.

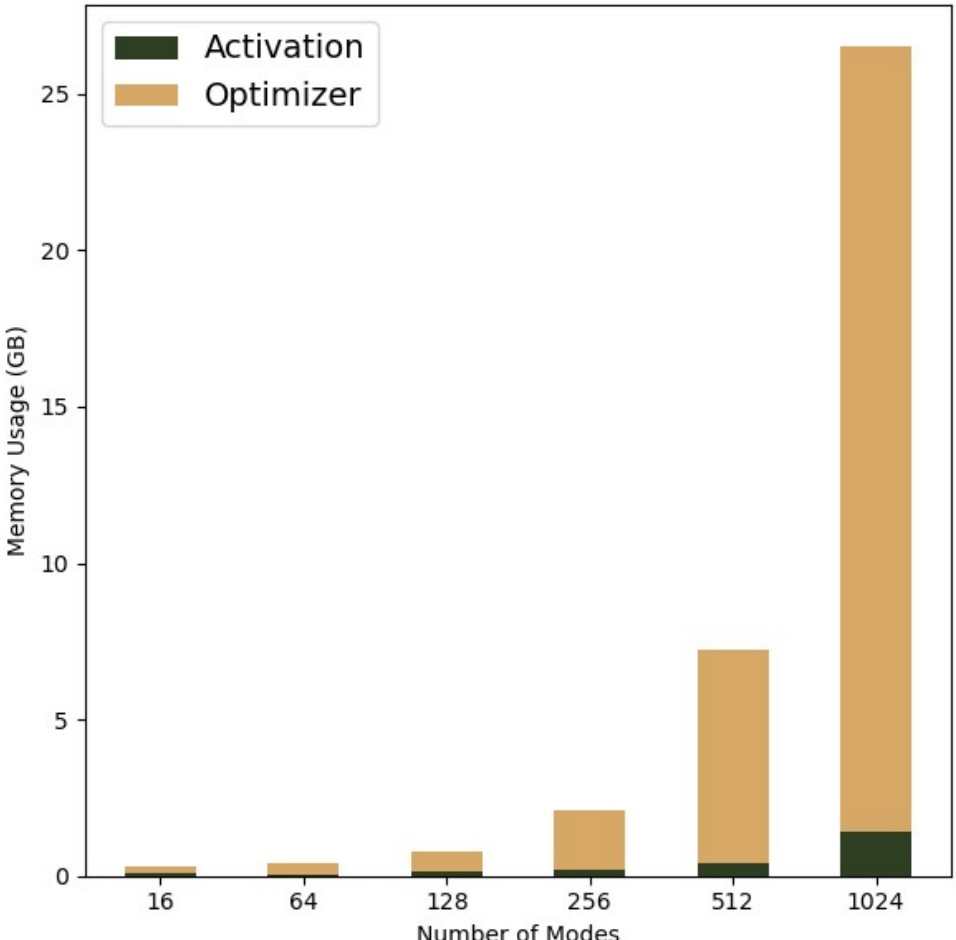

Figure 8: Memory usage in FNO as a function of the number of modes

As evident from the figure, the memory consumption is divided into two main categories: activation memory and optimizer memory. The activation memory, represented by the dark green bars, remains relatively constant and low across different numbers of modes. This stability in activation memory is a positive attribute of FNOs, indicating that the forward and backward passes do not significantly increase memory requirements as the model complexity grows.

However, the optimizer memory, shown in yellow, exhibits a dramatic increase as the number of modes grows. This exponential growth in optimizer memory becomes particularly pronounced for models with more than 128 modes. For instance, when the number of modes reaches 1024, the optimizer memory dominates the total memory usage, far exceeding the memory required for activations.

This trend highlights a critical bottleneck in scaling FNO models to higher resolutions or more complex problems. The optimizer's memory footprint, which includes storage for gradients, momentum, and adaptive learning rate parameters, becomes the primary limiting factor. This observation motivates the need for memory-efficient optimization techniques like TENSORGRAD, which specifically target the reduction of optimizer memory usage while maintaining model performance.

## F  Matrix GaLore

GaLore (Gradient Low-Rank Projection) (Zhao et al., 2024) is a memory-efficient optimization method designed to reduce the memory overhead of gradient updates by leveraging the low-rank structure often present in gradient matrices. Specifically, GaLore operates on weight matrices $W \in \mathbb{R}^{m \times n}$ and their corresponding gradient matrices $G \in \mathbb{R}^{m \times n}$. For a given rank $r$, GaLore computes the SVD of the gradient matrix every $T$ steps, forms projection matrices using the first $r$ singular vectors, then projects the gradient onto this low-rank subspace to perform optimization. After computing the optimizer update, the gradients are projected back to their full rank for use in the model. This approach allows GaLore to maintain a low memory footprint by storing and updating only the low-rank representations of gradients.

**Challenges of applying GaLore to neural operators.**  In order to apply standard GaLore to tensor weights, the weights must first be reshaped into a matrix to compute the SVD for projection into a low-rank space. GaLore takes one rank parameter, $r$, and projects high-rank gradients onto the first $r$ basis vectors of the corresponding SVD rotation matrix. When the weight matrix corresponds to an operator that maps between vectors, a single rank cutoff can be applied while preserving most information.

However, in the tensor case, weights correspond to higher-order maps between function spaces. Depending on the chosen strategy for reshaping tensor weights into a matrix, applying a single-dimension rank cutoff to the matrix may discard key information - for instance, for a tensor $W \in \mathbb{C}^{A \times B \times m \times m}$, where $A$ is the number of input channels, $B$ is the number of output channels, and $m$ is the number of truncated Fourier basis modes along each dimension, reshaping $W$ into $W' \in \mathbb{C}^{ABm \times m}$ and cutting off the first dimension at rank $r$ may remove all information about Fourier modes along the first dimension, making function learning impossible. We call this method *GaLore* and provide several comparisons to demonstrate its flaws.

One flaw is the **Loss of mode-specific information**: by collapsing multiple tensor dimensions into one matrix dimension, we lose the ability to preserve different amounts of information along each tensor mode. The other is that we have an **imbalanced projection**: Projecting only on one side of the reshaped matrix (e.g., only $U$ or only $V$ from the SVD) can severely limit the operator's capacity. However, projecting on both sides often leads to training instability and failure to converge. This method also encounters **rank selection issues**: Choosing a single rank cutoff for the reshaped matrix makes it difficult to balance information preservation across all the original tensor dimensions. A rank that preserves enough information for one dimension may be too restrictive for another.

## G  Tensor Decomposition

Tensors are multidimensional arrays that generalize the concepts of vectors (first-order tensors) and matrices (second-order tensors) to higher orders. An $N$th-order tensor $\mathcal{X} \in \mathbb{C}^{I_1 \times I_2 \times \cdots \times I_N}$ is an $N$-way array where

each mode $n$ has dimension $I_n$. Like matrices, in tensors, we can decompose the tensors into low-rank factors using the Tucker decomposition, also known as the higher-order SVD (HOSVD), which decomposes a tensor into a core tensor multiplied by a matrix along each mode:

$$\mathcal{X} \approx \mathcal{G} \times_1 U^{(1)} \times_2 U^{(2)} \cdots \times_N U^{(N)} = [\![\mathcal{G}; U^{(1)}, U^{(2)}, \ldots, U^{(N)}]\!] \tag{5}$$

where $\mathcal{G} \in \mathbb{C}^{R_1 \times R_2 \times \cdots \times R_N}$ is the core tensor, $U^{(n)} \in \mathbb{C}^{I_n \times R_n}$ are factor matrices, and $\times_n$ denotes the $n$-mode product. Two critical aspects of the Tucker decomposition make it particularly suitable for our TENSORGRAD method:

1. **Equivalence to SVD in 2D**: In the special case of 2D tensors (matrices), the Tucker decomposition reduces to the familiar SVD. The core tensor $\mathcal{G}$ becomes equivalent to the diagonal matrix $\Sigma$ in SVD, while the factor matrices correspond to the orthogonal matrices $U$ and $V$ (Kolda and Bader, 2009). This property ensures that our method seamlessly extends the principles of matrix-based techniques to higher-order tensors.

2. **Orthogonality of factor matrices**: The factor matrices $U^{(n)}$ in Tucker decomposition are orthogonal, mirroring the properties of $U$ and $V$ in SVD. This orthogonality is crucial for the efficiency and stability of the GaLore method. Specifically:

(a) *Projection efficiency*: The orthogonality allows us to project tensors onto the subspace spanned by these matrices through simple matrix multiplication, without the need for costly inverse computations.

(b) *Easy inversion*: When we need to reverse the projection, we can simply use the transpose of these orthogonal matrices instead of computing their inverses. This property is expressed mathematically as $(U^{(n)})^T U^{(n)} = I$, where $I$ is the identity matrix.

(c) *Numerical stability*: Orthogonal matrices have a condition number of 1, ensuring that the projection and its inverse are numerically stable operations, even for high-dimensional tensors.

Tensor decompositions such as Tucker have been extensively applied to deep learning architectures, but primarily for compressing or accelerating *weights* rather than *gradients*. Prior work includes applications of low-rank and tensorized models in convolutional networks, recurrent networks, transformers, 3D rendering, and operator learning (Phan et al., 2020; Wu et al., 2020; Astrid and Lee, 2017; Lee et al., 2021; Yin et al., 2021; Loeschcke et al., 2024b; Liu and Ng, 2022; Kossaifi et al., 2023; Novikov et al., 2015; Lebedev et al., 2015; Kim et al., 2016; Panagakis et al., 2021)

### G.1    Implementation

The **low-rank component** of TENSORGRAD is implemented using TensorLy (Kossaifi et al., 2019), which provides an efficient implementation of the Tucker decomposition based on Higher-Order Orthogonal Iteration (HOI). Given an input tensor $\mathcal{X}$, HOI approximates the factor matrices $U^{(n)}{}_n$ by computing truncated SVDs of mode-$n$ unfoldings and iteratively refines them to minimize the Frobenius reconstruction error. The decomposition supports warm restarts, allowing factors to be reused across steps to reduce iteration cost. All subsequent operations, like compressing the gradients and reconstructing the low-rank updates, are performed using PyTorch.

### G.2    Rank selection heuristic.

Since optimal multilinear Tucker ranks are NP-hard (Hillar and Lim, 2013) to determine, we use a memory-budget based heuristic. Given a tensor $\mathcal{G} \in \mathbb{R}^{n_1 \times n_2 \times n_3 \times n_4}$ and a desired compression ratio $\rho \in (0, 1)$ for the low-rank component, we select ranks $(r_1, r_2, r_3, r_4)$ such that:

$$\sum_{i=1}^{4} n_i r_i \leq \rho \cdot \prod_{i=1}^{4} n_i.$$

In practice, we use uniform rank percentages per mode unless otherwise specified.

## H    Structured Sparse Projections

Sparse projections provide an alternative to low-rank compression by selectively retaining a subset of gradient values. GRASS (Muhamed et al., 2024) introduced this approach for gradient matrices, where optimizer statistics are maintained only for a selected subset of rows, leading to substantial memory savings without requiring an SVD.

This idea can be directly extended to tensor gradients by applying structured sparsity, where the mask $\Omega$ selects entire slices along one or more tensor modes. For a gradient tensor $\mathcal{G} \in \mathbb{C}^{I_1 \times \cdots \times I_N}$, the structured mask is defined as a Cartesian product of per-mode index sets:

$$\Omega = M^{(1)} \times M^{(2)} \times \cdots \times M^{(N)},$$

and the projected tensor is $\hat{\mathcal{G}} = \mathcal{G}[\Omega]$. This form of sparsity is efficient, as the mask stores only $\sum_n r_n$ indices, and projection is implemented via multi-dimensional slicing.

The selection strategies presented in GRASS (Muhamed et al., 2024) can also be applied to tensors for constructing the sets $M^{(n)}$, including random sampling (Rand-$k$), and norm-based rules such as Top-$k$ and Prob-$k$. However, norm-based selection requires unfolding the tensor along each mode and computing slice norms $s^{(n)}i = \|\mathcal{G}i :: \cdots\|_2$, which may discard important structural information depending on the unfolding.

Structured sparsity is particularly effective when gradient mass is concentrated along specific tensor modes. However, it lacks flexibility in targeting scattered or irregularly positioned high-importance entries.

To restore the sparse tensor to its original shape during optimizer updates, we use a back-projection operation that places the retained values back into their original positions and fills all other entries with zeros. This is implemented using the adjoint of the projection operator:

$$\tilde{\mathcal{G}} = \mathcal{P}\Omega^\top(\hat{\mathbf{g}}) =, \mathrm{unvec}\left(P\Omega^\top \hat{\mathbf{g}}\right),$$

where $\hat{\mathbf{g}} = P_\Omega \mathrm{vec}(\mathcal{G})$ is the compressed vector of retained entries.

## I    Mixed Precision.

Our empirical findings are consistent with the mixed-precision analysis of Tu et al. (2024) who show that reduced-precision errors in FNOs can be bounded by discretization errors. We go one step further by storing both weights and gradients in half precision while keeping optimizer states in full precision. When we quantize the optimizer states themselves to half precision (Mixed-2), performance degrades substantially (Tab. 3), likely because the limited mantissa cannot represent the small second-moment values (often around $10^{-5}$).

## J    Profiling Methodology

To analyze the performance and memory usage of our TENSORGRAD method, we implemented a comprehensive profiling setup using PyTorch's built-in profiler. This allowed us to gain detailed insights into the computational and memory requirements of our algorithm compared to baseline methods.

**Detailed Memory Breakdown.** We implemented a detailed memory tracking system to break GPU memory usage down into key categories.

PyTorch provides access to a memory profiler, which collects granular information about each block of memory allocated on the CPU and GPU within the context window in which it is invoked. The profiler is run over a set number of iterations, each of which corresponds to a forward and backward pass through the model and one step of the optimizer. The profiler discards data from the first iteration due to the additional overhead of initialization and allocation, as well as a specified number of warmup steps. The profiler collects data for a set number of iterations, and averages this process over a series of repeats. In our experiments, we discarded 1 step, warmed up for 1 step, collected data for 3 steps, and performed 3 repeats.

To break down memory by category, we relied on the automatic categorization utility provided by the profiler's Memory Timeline, which coalesces all individual blocks into one of eight categories, which we enumerate and explain below.

- **Model parameters**: Tensors registered as instances of `torch.nn.Parameter`.

- **Optimizer states:** Optimizer states are tensors stored within the optimizer that are used to compute the final gradient. For Adam optimizer, this includes first and second moment estimates, which are each a tensor of the same size as the gradients themselves.

- **Inputs:** memory allocations that occur during data loading and preprocessing.

- **Activations:** Temporary tensors created during the forward pass of the model.

- **Gradients:** The tensors added to model parameters during the optimizer step. They are computed for each parameter by backpropagating loss through the model.

- **Autograd Detail:** Extra memory allocated during PyTorch's autograd operation, including memory used for storing computational graphs and intermediate results needed for backpropagation.

- **Temporary:** Temporary Buffers are short-lived tensors that are created and destroyed within a single operation or a small set of operations. For method, it is often used in complex computations like FFTs or tensor decompositions within galore.

- **"None":** PyTorch's profiler is often unable to categorize memory immediately upon allocation. In our tests, we compared figures generated by coalescing a profiler's memory timeline one-to-one with more granular block-level traces acquired from PyTorch's CUDA memory snapshot to conclude that memory tagged as None is either eventually reclassified as another type, or corresponds to an intermediate activation that is deallocated during backpropagation. For this reason, we chose to classify "None" memory "Intermediate". For simplicity in the final figure, we also included temporary buffers, autograd detail buffers and activations in the Intermediate category, though our profiler maintains the capability to collect data for each category separately.

To accurately break down the profiler's memory timeline into these categories, we took the following approach. First, we noticed that the peak memory allocation often occurred at a step when the total of None memory was very high, and that totals reported during that timestep for other categories were zero or very low. To obtain informative breakdowns for those categories, informed by our analysis of memory tagged as None, we identified the timestep at which peak memory usage occurred *after excluding None-tagged memory from the total*. This allowed us to break down all non-intermediate memory by its true classification once properly tagged. Once we obtained this breakdown, we subtracted the total from the true total memory allocation observed at the peak to obtain the true amount of intermediate memory allocated at peak.

## K Additional Results

This section provides extended experimental results demonstrating the effectiveness and generality of TEN-SORGRAD across a wide range of PDE datasets, architectures, and dimensional settings. In addition to standard 2D benchmarks such as Burgers, Darcy flow, Navier–Stokes, and Electromagnetic wave propagation, we also include results on the 3D ShapeNet Car benchmark using the Geometry-Informed Neural Operator (GINO), highlighting that TENSORGRAD extends beyond FNO-based architectures.

Overall, TENSORGRAD consistently achieves strong memory–accuracy trade-offs. For Burgers, performance improves steadily with increasing rank. On Darcy flow, TENSORGRAD achieves up to a 49% gain in test loss at rank 0.25 while reducing optimizer memory by 76%. Navier–Stokes experiments demonstrate that TENSORGRAD maintains comparable performance at lower ranks while significantly reducing memory usage, and Electromagnetic simulations show improvements of up to 11%.

---

**Algorithm 2** GaLore

---

**Require:** A layer weight tensor $\mathcal{W} \in \mathbb{C}^{N_1 \times N_2 \times N_3 \times N_4}$. Step size $\eta$, scale factor $\alpha$, decay rates $\beta_1, \beta_2$, rank $r$, subspace change frequency $T$, chosen dimension $d$.
 1: Initialize first-order moment $\mathcal{M}_0 \in \mathbb{C}^{r \times N_2 \times N_3 \times N_4} \leftarrow 0$ (Assuming matrization 1)
 2: Initialize second-order moment $\mathcal{V}_0 \in \mathbb{R}^{r \times N_2 \times N_3 \times N_4} \leftarrow 0$ (Assuming matrization 1)
 3: Initialize step $t \leftarrow 0$
 4: **repeat**
 5:     $\mathcal{G}_t \in \mathbb{C}^{N_1 \times N_2 \times N_3 \times N_4} \leftarrow -\nabla_{\mathcal{W}} \phi_t(\mathcal{W}_t)$
 6:     $G_t^{(d)} \leftarrow \text{Reshape}(\mathcal{G}_t, (N_d, \prod_{i \neq d} N_i))$                    ▷ Reshape tensor to matrix
 7:     **if** $t \bmod T = 0$ **then**
 8:         $U, \Sigma, V^\top \leftarrow \text{SVD}(G_t^{(d)})$                         ▷ Compute SVD
 9:         $P \leftarrow V[:, : r]^\top$                           ▷ Select $r$ right singular vectors
10:     **end if**
11:     $R_t \leftarrow G_t^{(d)} P^\top$                               ▷ Project gradient into compact space
12:     **UPDATE**$(R_t)$ by Adam:
13:         $M_t \leftarrow \beta_1 \cdot M_{t-1} + (1 - \beta_1) \cdot R_t$
14:         $V_t \leftarrow \beta_2 \cdot V_{t-1} + (1 - \beta_2) \cdot |R_t|^2$
15:         $M_t \leftarrow M_t / (1 - \beta_1^t)$
16:         $V_t \leftarrow V_t / (1 - \beta_2^t)$
17:         $N_t \leftarrow M_t / (\sqrt{V_t} + \epsilon)$
18:     $\tilde{G}_t^{(d)} \leftarrow \alpha \cdot N_t P$                            ▷ Project back to original space
19:     $\tilde{\mathcal{G}}_t \leftarrow \text{Reshape}(\tilde{G}_t^{(d)}, (N_1, N_2, N_3, N_4))$            ▷ Reshape back to tensor
20:     $\mathcal{W}_t \leftarrow \mathcal{W}_{t-1} + \eta \cdot \tilde{\mathcal{G}}_t$
21:     $t \leftarrow t + 1$
22: **until** convergence criteria met
23: **return** $\mathcal{W}_t$

---

### K.1 Compatibility with TFNO (factorized weights).

Tensorized FNO (TFNO) (Kossaifi et al., 2023) replaces the standard dense convolutional weight tensors in FNO with Tucker-factorized representations, decomposing each weight tensor into low-rank mode factors and a small core tensor to reduce parameter count and model memory while preserving the operator's expressive capacity. Importantly, TFNO constrains the *weights* to remain low-rank throughout training, whereas TENSORGRAD instead targets only the *optimizer states*.

We evaluate TENSORGRAD on top of TFNO at varying weight ranks on Navier-Stokes $128 \times 128$ and $1024 \times 1024$. As shown in Tab. 5, TENSORGRAD closely matches Adam across all TFNO ranks, demonstrating that optimizer-state compression remains effective even when the weights themselves are factorized. Here, the *baseline* corresponds to the standard FNO trained with full-rank weights (no TFNO factorization), using the same architecture, optimizer, and training schedule.

These results show that TENSORGRAD integrates with TFNO, preserving performance across weight ranks while reducing optimizer memory, and highlight the orthogonality between low-rank structure in weights and low-rank structure in gradients. Importantly, the performance difference introduced by TENSORGRAD alone is smaller than that caused by TFNO weight factorization. On NS 1024, the baseline increases from 20.08 to 20.24 with TENSORGRAD, whereas TFNO at 50% rank increases the error to 22.12, indicating that most of the degradation originates from the weight factorization rather than the compressed optimizer states.

### K.2 3D ShapeNet with Geometry-Informed Neural Operator.

In Tab. 6, we show that TENSORGRAD also applies to 3D problems and architectures beyond FNO. We evaluate it on the 3D ShapeNet Car (Chang et al., 2015) benchmark using the Geometry-Informed Neural

Table 5: TENSORGRAD with tensorized weights (TFNO) on Navier–Stokes at two resolutions. We report L2 test loss; TFNO rank denotes retained weight percentage. NS 128 uses Re $\approx 10^3$ (500 steps) and NS 1024 uses Re $\approx 10^5$ (50 steps).

| TFNO Rank | NS 128 ($\times 10^{-3}$) | | NS 1024 ($\times 10^{-2}$) | |
|---|---|---|---|---|
| | Adam | TensorGRaD | Adam | TensorGRaD |
| 15% | 5.787 | 5.821 | 24.09 | 24.21 |
| 25% | 5.672 | 5.729 | 23.13 | 23.17 |
| 50% | 5.912 | 5.967 | 22.12 | 22.43 |
| Baseline | 5.660 | 5.720 | 20.08 | 20.24 |

Table 6: Performance on the 3D ShapeNet Car (Chang et al., 2015) benchmark using 3D GINO. We report relative train and test errors (%). TensorGRaD uses 20% low-rank + 5% sparse optimizer states.

| Metric | Adam | TensorGRaD |
|---|---|---|
| Train Error (%) | 2.94 | 3.37 |
| Test Error (%) | 8.53 | 8.74 |

Operator (GINO) (Li et al., 2023), which lifts mesh data to a 3D Eulerian grid, mixes features on the grid, and projects predictions back to the mesh. Using the official training setup, we replace only the optimizer with TENSORGRAD (20% low-rank + 5% sparse). TENSORGRAD closely matches Adam's performance while reducing optimizer memory by up to 75%, confirming its applicability to 3D data and architectures beyond FNO.

Table 7: Full results for memory and accuracy comparison on Navier–Stokes $1024 \times 1024$ with Reynolds number $2 \times 10^5$. Train and test losses are $L_2 \times 10^{-2}$ (mean $\pm$ 1 standard error over three seeds). "Mixed" uses half-precision weights and gradients with a mixed-precision forward pass. Memory is a rounded peak GPU allocation.

| Model | Memory (GB) | Precision | Train $L_2$ | Test $L_2$ |
|---|---|---|---|---|
| **Low-Rank Only** | | | | |
| 25% | 46 | Full | $5.37 \pm 0.08$ | $17.19 \pm 0.23$ |
| | 29 | Mixed | $6.92 \pm 0.19$ | $17.09 \pm 0.19$ |
| 25% rand-svd | 46 | Full | $5.93 \pm 0.36$ | $17.23 \pm 0.21$ |
| 50% Matrix (d=1) | 49 | Full | $29.63 \pm 1.46$ | $33.11 \pm 1.21$ |
| **Sparse Only** | | | | |
| 25% | 46 | Full | $6.39 \pm 0.32$ | $18.73 \pm 0.08$ |
| | 29 | Mixed | $7.37 \pm 0.14$ | $18.54 \pm 0.34$ |
| 50% | 49 | Full | $5.35 \pm 0.08$ | $17.21 \pm 0.16$ |
| | 32 | Mixed | $6.02 \pm 0.28$ | $17.38 \pm 0.31$ |
| **TensorGRaD** | | | | |
| 25% (5+20) | 46 | Full | $5.36 \pm 0.05$ | $16.82 \pm 0.18$ |
| | 29 | Mixed | $6.42 \pm 0.15$ | $16.87 \pm 0.15$ |
| **Adam Baseline** | | | | |
| No Compression | 52 | Full | $3.94 \pm 0.22$ | $17.02 \pm 0.18$ |
| | 37 | Mixed | $4.86 \pm 0.26$ | $17.01 \pm 0.19$ |

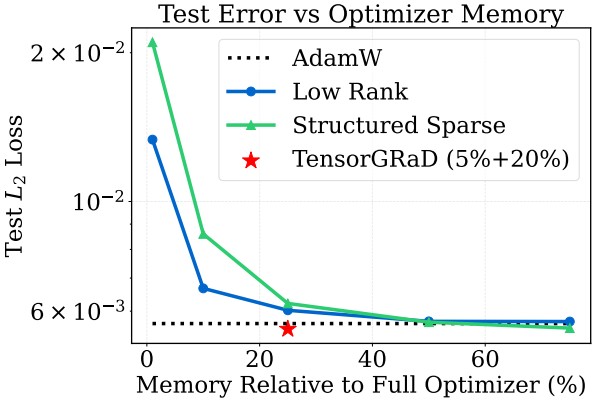 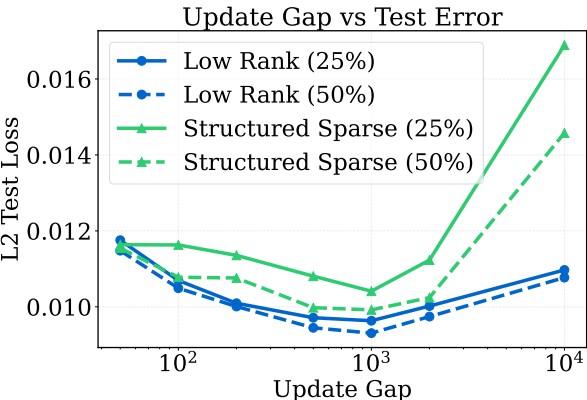

Figure 9: Left: Low-rank, structured, at various rank percentages vs TENSORGRAD at 25% and baseline Adam optimizer. Right: Effect of projection gap on $L_2$ test loss for sparse and low-rank projections at 25% and 50% compression, trained for 150 epochs on the NS128 dataset.

Table 8: Model performance on Darcy-flow.

| Model | Test Loss (1e-2) at Rank Ratio | | | | | | Gain (%) |
|---|---|---|---|---|---|---|---|
| | 0.01 | 0.1 | 0.25 | 0.5 | 0.75 | 1.0 | |
| FNO Baseline | - | - | - | - | - | 0.205 | / |
| FNO - Tensor Low-rank | **0.147** | **0.108** | **0.105** | **0.107** | **0.140** | **0.173** | **49** |
| FNO - GaLore (d=1) | 0.256 | 0.232 | 0.212 | 0.245 | 0.201 | 0.190 | 8 |
| FNO - GaLore (d=2) | 0.203 | 0.192 | 0.168 | 0.178 | 0.170 | 0.180 | 19 |
| FNO - GaLore (d=3) | 0.234 | 0.212 | 0.201 | 0.193 | 0.196 | 0.182 | 11 |

### K.3 Update frequency of tensor decomposition ablation.

Fig. 9 illustrates how the projection gap affects TENSORGRAD on the NS128 task over the first 150 epochs, comparing 25% and 50% rank settings. In line with findings from (Zhao et al., 2024; Muhamed et al., 2024), increasing the projection gap can improve generalization, as switching subspaces introduces a small amount of noise that can harm the performance, if done too frequently

We observe that low-rank projections are less sensitive to the projection gap and tend to perform better in the early stages of training. In our experiments, they consistently lead during roughly the first half of training, after which structured sparsity gradually catches up. A likely explanation is that low-rank projections quickly capture the dominant modes of the gradient with fewer parameters, enabling faster initial convergence. In contrast, structured sparsity requires more iterations to recover the same signal, as it discards parts of the gradient during each step. This slower adaptation may, however, contribute to better generalization by reducing overfitting in later epochs.

### K.4 Tensor Low-rank vs matrix GaLore

We highlight that our higher-order low-rank factorization version of TENSORGRAD shows superior performance to the direct extension of GaLore as described in F. In Tab. 8, we show results for Darcy Flow comparing a higher-order low-rank factorization to construct the projection matrix instead of naively unfolding the tensor to 2d and removing the inherent structure. Comparing different unfoldings (variations of $d$), we observed up to a 48% improvement in test loss with a rank of 0.25, while reducing the optimizer state memory from 2.09GB to 0.5GB.

We evaluate three approaches to matricizing a tensor gradient with shape $C_{in} \times C_{out} \times M_x \times M_y$. The first, which we call "rollout=1", combines the last 3 dimensions into one matrix dimension, resulting in a matrix of shape $C_{in} \times (C_{out} * M_x * M_y)$. The second, "rollout=2", combines the first two dimensions into

Table 9: Evaluating low-rank tensor factorization across various tasks.

| Model | Rank Ratio | Memory (GB) | Train (Loss ($\times 10^{-2}$)) | Test $H_1$ (Loss ($\times 10^{-2}$)) | Test $L_2$ (Loss ($\times 10^{-2}$)) | Gain (%) |
|---|---|---|---|---|---|---|
| **Darcy** | | | | | | |
| Baseline | 1.0 | 8.88 | 0.7151 | 1.6230 | 0.2050 | / |
| GaLore (d=2) | 0.25 | 7.34 | 0.4200 | 1.3210 | 0.1680 | 19 |
| **Tensor low-rank** | 0.25 | 7.32 | **0.2930** | **0.8680** | **0.1050** | **48.8** |
| **Navier-Stokes** | | | | | | |
| Baseline | 1.0 | 77 | 1.0630 | 1.9010 | 0.6152 | / |
| GaLore (d=1) | 0.5 | 68 | 4.3340 | 5.5830 | 1.9952 | -223 |
| **Tensor low-rank** | 0.5 | 55 | 1.2340 | 2.0850 | 0.6480 | -5.4 |
| **ElectroMagnetic** | | | | | | |
| Baseline | 1.0 | 4.83 | 2.973 | 0.1902 | 0.2000 | / |
| GaLore (d=2) | 0.25 | 4.83 | 2.392 | 0.1802 | 0.1900 | 5 |
| **Tensor low-rank** | 0.25 | 4.63 | **2.132** | **0.1681** | **0.1782** | **11** |
| **Burgers** | | | | | | |
| Baseline | 1.0 | 3.94 | 0.0064 | 0.0050 | 0.0026 | / |
| GaLore (d=2) | 0.5 | 3.88 | 0.0052 | 0.0100 | 0.0062 | -138 |
| **Tensor low-rank** | 0.5 | 3.87 | **0.0026** | **0.0041** | **0.0025** | **+5** |

the first matrix dimension and the last two dimensions into the second matrix dimension, resulting in a matrix of shape $(C_{in} * C_{out}) \times (M_x * M_y)$. The last, "rollout=3", combines the last three dimensions into the second matrix dimension, resulting in a matrix of shape $C_{in} \times (C_{out} * M_x * M_y)$. We showcase results and comparisons for all three approaches in Tab. 13.

All of the subsequent results are with varying rank ratios on the TensorGrad method for all datasets. We report both the training and testing loss/accuracy.

Table 10: Model performance on Burgers

| Model | Rank Ratio | Train Loss (1e-4) | Test Loss(1e-4) | Gain (%) |
|---|---|---|---|---|
| FNO Baseline | Full Rank | 0.205 | 0.262 | / |
| FNO - **Tensor low-rank** | 0.1 | 0.115 | 0.321 | -19 |
| FNO - **Tensor low-rank** | 0.25 | 0.095 | 0.271 | -4 |
| FNO - **Tensor low-rank** | 0.5 | 0.086 | 0.253 | +5 |
| FNO - **Tensor low-rank** | 0.75 | 0.083 | 0.246 | +8 |
| FNO - **Tensor low-rank** | 1.00 | 0.083 | **0.242** | **+9** |

Table 11: Model performance on Darcy-flow

| Model | Rank Ratio | Train Loss (1e-2) | Test Loss(1e-2) | Gain (%) |
|---|---|---|---|---|
| FNO Baseline | Full Rank | 0.715 | 0.205 | / |
| FNO - **Tensor low-rank** | 0.01 | 0.465 | 0.147 | +30 |
| FNO - **Tensor low-rank** | 0.1 | 0.323 | 0.108 | +48 |
| FNO - **Tensor low-rank** | 0.25 | **0.293** | **0.105** | **+49** |
| FNO - **Tensor low-rank** | 0.5 | 0.275 | 0.107 | +49 |
| FNO - **Tensor low-rank** | 0.75 | 0.379 | 0.140 | +32 |
| FNO - **Tensor low-rank** | 1.00 | 0.715 | 0.173 | +16 |

Table 12: Model performance on EM.

| Model | Test Loss (1e-2) at Rank Ratio | | | | | | Gain (%) |
|---|---|---|---|---|---|---|---|
| | 0.01 | 0.1 | 0.25 | 0.5 | 0.75 | 1.0 | |
| FNO Baseline | - | - | - | - | - | 0.200 | / |
| FNO - **Tensor low-rank** | 0.187 | 0.185 | **0.178** | 0.176 | 0.174 | 0.206 | **11** |
| FNO - GaLore (d=1) | 0.213 | 0.192 | 0.193 | 0.189 | 0.194 | 0.200 | 7 |
| FNO - GaLore (d=2) | 0.205 | 0.206 | 0.195 | 0.196 | 0.201 | 0.199 | 3 |

Table 13: Ablation: GaLore and **Tensor low-rank** Rank Comparison

| Method | % orig. parameters | GaLore Test $L_2$ ($\times 10^{-2}$) | **Tensor low-rank** Test $L_2$ ($\times 10^{-2}$) |
|---|---|---|---|
| GaLore (d=1) | 25 | $2.495 \pm 0.920$ | **0.9141** $\pm 0.0064$ |
| | 50 | $3.594 \pm 0.885$ | **0.7622** $\pm 0.0984$ |
| | 75 | $3.298 \pm 1.96$ | **0.6697** $\pm 0.0746$ |
| GaLore (d=2) | 25 | $8.715 \pm 0.252$ | **0.9141** $\pm 0.0064$ |
| | 50 | $8.683 \pm 0.0014$ | **0.7622** $\pm 0.0984$ |
| | 75 | $8.950 \pm 0.0141$ | **0.6697** $\pm 0.0746$ |
| GaLore (d=3) | 25 | $8.723 \pm 0.0149$ | **0.9141** $\pm 0.0064$ |
| | 50 | $8.702 \pm 0.0108$ | **0.7622** $\pm 0.0984$ |
| | 75 | $8.585 \pm 0.0171$ | **0.6697** $\pm 0.0746$ |

### K.5 Structured Sparsity Mask Strategy

Tab. 14 shows an ablation of three structured sparse projection strategies: **Top-$k$**, **Probability**, and **Rand-$k$**, evaluated at 25% and 50% sparsity.

We find that **Rand-$k$** consistently outperforms both Top-$k$ and Probability sampling, especially at higher sparsity (50%). This aligns with prior findings in GRASS (Muhamed et al., 2024).

The performance gap stems from how each method selects entries:

- **Top-$k$** selects slices with the highest norms.

- **Probability** sampling assigns selection probabilities proportional to slice norms.

- **Rand-$k$** uniformly samples slices at random, without relying on norm computations.

Table 14: Ablation of sparse projection methods at 25% and 50% sparsity. Train and test losses are reported as $L_2$ loss ($\times 10^{-3}$) with mean ± std over multiple runs.

| Method | Train loss | Test $L_2$ |
|---|---|---|
| **Top-$k$** | | |
| 25% | 12.05 | 6.50 |
| 50% | 12.92 | 7.93 |
| **Probability** | | |
| 25% | 10.91 | 6.20 |
| 50% | 9.59 | 6.07 |
| **Rand-$k$** | | |
| 25% | 12.01 | 6.22 |
| 50% | **9.14** | **5.54** |

Table 15: Model performance on EM

| Model | Rank Ratio | Train Loss | Test Loss | Gain (%) |
|---|---|---|---|---|
| Complex FNO Baseline | Full Rank | 2.973 | 0.200 | / |
| Complex FNO - **Tensor low-rank** | 0.01 | 4.198 | 0.249 | -20 |
| Complex FNO - **Tensor low-rank** | 0.1 | 2.936 | 0.217 | -8 |
| Complex FNO - **Tensor low-rank** | 0.25 | **2.132** | **0.178** | +11 |
| Complex FNO - **Tensor low-rank** | 0.5 | 2.430 | 0.184 | +8 |
| Complex FNO - **Tensor low-rank** | 0.75 | 2.719 | 0.192 | +4 |
| Complex FNO - **Tensor low-rank** | 1.00 | 2.397 | 0.185 | +8 |

## L  Architecture and Training Details

**Sobolev Loss for PDE Training.**  In training NOs for PDEs we employ both the $L^2$ and Sobolev $H^1$ losses to provide a comprehensive assessment of model performance. While the $L^2$ loss measures point-wise accuracy of predictions, the $H^1$ loss, defined as $\|u - \hat{u}\|^2_{H^1} = \|u - \hat{u}\|^2_{L^2} + \|\nabla u - \nabla \hat{u}\|^2_{L^2}$, accounts for both the function values and their gradients. This is particularly crucial for PDEs, as it ensures that the learned solutions not only match the target values but also preserve the smoothness and differential properties inherent in the physical systems being modeled.

**Sobolev Loss for Complex Wave Phenomena.**  The Sobolev $H^1$ loss proves especially valuable when dealing with complex wave phenomena, as demonstrated in our experiments with the EM Dataset using Complex-FNOs. In this case, the $H^1$ loss not only measures the accuracy of the predicted complex electric field envelope but also ensures that its spatial derivatives are correctly captured. This is crucial for accurately representing the rapid oscillations and sharp peaks characteristic of EM waves. Our results show that **Tensor low-rank** with a rank ratio of 0.25 achieved an 11% improvement in overall test loss compared to the baseline, with the $H^1$ loss decreasing from 0.1902 to 0.1681. This improvement is particularly significant given the challenging nature of the EM dataset, which involves predicting the complex electric field envelope resulting from nonlinear interactions in waveguides. The enhanced performance in $H^1$ loss indicates that our model not only matches the amplitude of the EM waves more accurately but also better captures the rapid spatial variations and peak formations. This is critical in applications such as optical pulse propagation, where precise modeling of field gradients and peak intensities is essential for predicting phenomena like second-harmonic generation and phase matching.

## M  Slowdown in Training

In this section we quantify the runtime overhead introduced by TENSORGRAD. Tab. 17 reports the per-epoch wall-clock time measured on an A100 with batch size 32 for AdamW, tensor low-rank only, the sparse+low-rank variant (TENSORGRAD), and matricized GaLore. All experiments use the same hardware and batch configuration. We use update frequency 1000 for this experiment.

The dominant overhead arises from the Tucker decomposition step; the sparse projection adds only a minor additional cost. Because the decomposition is recomputed every $T$ steps, its cost is amortized, resulting in a modest 5–25% slowdown depending on the chosen rank and update frequency.

Despite this overhead, TENSORGRAD reduces optimizer memory by up to 75%, which in practice allows training at higher resolutions or with larger batch sizes on the same hardware. We also note that TENSOR-GRAD is slightly slower than tensor low-rank alone due to the additional sparse projection, but both remain within a relatively small overhead range.

Moreover, we have incorporated techniques such as "warm-restart" initialization of the tensor decomposition to amortize the computational overhead across training iterations. This helps minimize the impact on the overall training efficiency. We have also explored opportunities to further optimize the tensor decomposition computations, which could potentially reduce the training time slowdown even further.

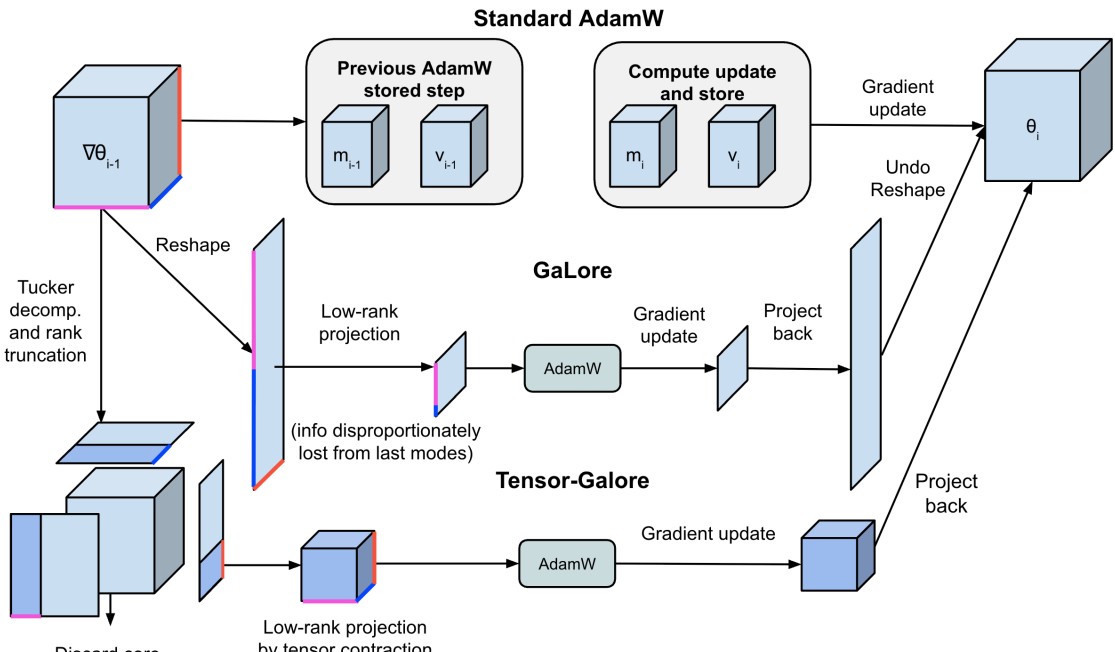

Figure 10: Comparison of the higher-order low-rank decomposition Tensor Low-Rank with standard Adam and GaLore. GaLore applies matrix-based low-rank projection after reshaping tensors. Our **Tensor low-rank** method leverages tensor decomposition to perform low-rank projection directly on tensor gradients, preserving multidimensional structure.

**Remark 2 (Real-Valued Analysis)** For clarity of presentation, we develop the theory of **Tensor low-rank** assuming all tensors are real-valued, i.e., $\mathcal{W}_l, \mathcal{G}_t \in \mathbb{R}^{N_1 \times N_2 \times N_3 \times N_4}$ and all associated operations are in real space. This simplification allows us to focus on the core geometric and algebraic properties without the additional complexity of complex conjugates and Hermitian operations. The extension to complex-valued tensors (as needed for Fourier Neural Operators where weights may be complex in the frequency domain) is straightforward: inner products become Hermitian inner products, transposes become conjugate transposes, and orthogonality conditions incorporate complex conjugates. All main results remain valid with these natural modifications.

# N   Parameter Complexity Analysis

To understand the theoretical advantages of **Tensor low-rank** over matrix-based GaLore, we provide a detailed analysis of the parameter complexity for both approaches. This analysis demonstrates why tensor decomposition leads to more efficient memory usage while maintaining expressiveness.

## N.1   Memory Analysis

We provide a theoretical analysis of the memory requirements for **Tensor low-rank** compared to baseline methods and matrix GaLore variants. Consider a weight tensor $W \in \mathbb{C}^{N_1 \times N_2 \times N_3 \times N_4}$ in a FNO Spectral layer. Tab. 18 summarizes the memory requirements for different methods. The baseline approach stores the full tensor and its corresponding optimizer states. For a rank ratio $r$ ($0 < r \leq 1$), **Tensor low-rank** requires storing the factor matrices, resulting in substantial memory savings, especially for the optimizer states. In this table, we assume the use of a complex-valued Adam optimizer, which typically requires two additional tensors (first and second moments) for each parameter.

---

**Algorithm 3** Adam with low-rank tensor decomposition

---

**Require:** A layer weight tensor $\mathcal{W} \in \mathbb{C}^{N_1 \times N_2 \times N_3 \times N_4}$. Step size $\eta$, scale factor $\alpha$, decay rates $\beta_1, \beta_2$, rank $r$, subspace change frequency $T$.

1: Initialize first-order moment $\mathcal{M}_0 \in \mathbb{C}^{r \times r \times r \times r} \leftarrow 0$
2: Initialize second-order moment $\mathcal{V}_0 \in \mathbb{C}^{r \times r \times r \times r} \leftarrow 0$
3: Initialize step $t \leftarrow 0$
4: **repeat**
5:     $\mathcal{G}_t \in \mathbb{C}^{N_1 \times N_2 \times N_3 \times N_4} \leftarrow -\nabla_{\mathcal{W}} \phi_t(\mathcal{W}_t)$
6:     **if** $t \bmod T = 0$ **then**
7:         $\mathcal{C}, \{U^{(n)}\}_{n=1}^4 \leftarrow \text{Projection}(\mathcal{G}_t, \text{rank} = r)$
8:     **else**
9:         $\mathcal{C}, \{U^{(n)}\}_{n=1}^4 \leftarrow \mathcal{C}_{t-1}, \{U_{t-1}^{(n)}\}_{n=1}^4$         $\triangleright$ Reuse the previous projector.
10:     **end if**
11:     $\mathcal{R}_t \leftarrow \mathcal{G}_t \times_1 U^{(1)^\top} \times_2 U^{(2)^\top} \times_3 U^{(3)^\top} \times_4 U^{(4)^\top}$     $\triangleright$ Project gradient into compact space.
12:     **UPDATE**$(\mathcal{R}_t)$ by Adam:
13:         $\mathcal{M}_t \leftarrow \beta_1 \cdot \mathcal{M}_{t-1} + (1 - \beta_1) \cdot \mathcal{R}_t$
14:         $\mathcal{V}_t \leftarrow \beta_2 \cdot \mathcal{V}_{t-1} + (1 - \beta_2) \cdot |\mathcal{R}_t \overline{\mathcal{R}_t}|$     $\triangleright$ We use the complex conjugate update.
15:         $\mathcal{M}_t \leftarrow \mathcal{M}_t / (1 - \beta_1^t)$
16:         $\mathcal{V}_t \leftarrow \mathcal{V}_t / (1 - \beta_2^t)$
17:         $\mathcal{N}_t \leftarrow \mathcal{M}_t / (\sqrt{\mathcal{V}_t} + \epsilon)$
18:     $\tilde{\mathcal{G}}_t \leftarrow \alpha \cdot \mathcal{N}_t \times_1 U^{(1)} \times_2 U^{(2)} \times_3 U^{(3)} \times_4 U^{(4)}$     $\triangleright$ Project back to original space.
19:     $\mathcal{W}_t \leftarrow \mathcal{W}_{t-1} + \eta \cdot \tilde{\mathcal{G}}_t$
20:     $t \leftarrow t + 1$
21: **until** convergence criteria met.
22: **return** $\mathcal{W}_t$

---

### N.1.1 Memory analysis: matricized GaLore vs tensor low-rank.

Consider a 4D FNO spectral weight tensor $\mathcal{W} \in \mathbb{R}^{I_1 \times I_2 \times I_3 \times I_4}$, where $(I_1, I_2)$ are input/output channels and $(I_3, I_4)$ are Fourier modes. A direct matrix GaLore extension first matricizes $\mathcal{W}$, e.g. $\mathbf{W}_{(1)} \in \mathbb{R}^{I_1 \times (I_2 I_3 I_4)}$ (and permutations) or $\mathbf{W}_{(12)} \in \mathbb{R}^{(I_1 I_2) \times (I_3 I_4)}$, and then applies a rank-$R$ SVD. Storing the low-rank factors (ignoring the lower-order $+O(R)$ singular values) costs $P_{(1)} \approx R(I_1 + I_2 I_3 I_4)$ or $P_{(12)} \approx R(I_1 I_2 + I_3 I_4)$ parameters, so the memory scales with products of tensor mode sizes once the tensor is flattened.

In tensor low-rank, we instead use a Tucker decomposition with ranks $(R_1, R_2, R_3, R_4)$: $\mathcal{W} \approx \mathcal{G} \times_1 U^{(1)} \times_2 U^{(2)} \times_3 U^{(3)} \times_4 U^{(4)}$, where $\mathcal{G} \in \mathbb{R}^{R_1 \times R_2 \times R_3 \times R_4}$ and $U^{(n)} \in \mathbb{R}^{I_n \times R_n}$. The storage is $P_{\text{Tucker}} = R_1 R_2 R_3 R_4 + \sum_{n=1}^4 I_n R_n$, which preserves mode-wise structure and replaces multiplicative scaling with additive mode-wise terms plus a small core.

For a symmetric layer with $I_1 = I_2 = N$ and $I_3 = I_4 = M$, using equal Tucker ranks $R_1 = \cdots = R_4 = r$ and a matched matrix rank $R = r^2$ for the best-case unfolding $\mathbf{W}_{(12)} \in \mathbb{R}^{N^2 \times M^2}$, we obtain $P_{\text{Matrix}} \approx r^2(N^2 + M^2)$ versus $P_{\text{Tensor}} = r^4 + 2rN + 2rM$. For example, with $N = 64$, $M = 128$, and $r = 16$, $P_{\text{Matrix}} \approx 5{,}242{,}880$ while $P_{\text{Tensor}} = 71{,}680$, a $\approx 73\times$ reduction in stored scalars. This gap increases further as the mode resolution $M$ grows, which matches the high-resolution neural operator regime where optimizer-state memory dominates.

## O Tensor Operations and Notation

In this appendix we fix the tensor notation used in the theory. The main point is simple: a tensor gradient has several natural matrix views, one per mode. A GaLore-style method chooses one matrix view after flattening, while TENSORGRAD keeps the modes separate and projects each of them directly. The definitions below make this distinction precise.

**Definition 1 (Tensor)** An order-$d$ tensor $\mathcal{A} \in \mathbb{R}^{I_1 \times \cdots \times I_d}$ is a $d$-way array with entries $a_{i_1, \ldots, i_d}$, where $1 \le i_k \le I_k$ for each mode $k = 1, \ldots, d$.

**Definition 2 (Mode-$k$ unfolding)** The mode-$k$ unfolding of $\mathcal{A}$ is the matrix $\mathcal{A}_{(k)} \in \mathbb{R}^{I_k \times \prod_{m \neq k} I_m}$ obtained by arranging all mode-$k$ fibers as columns. Equivalently,

$$(\mathcal{A}_{(k)})_{i_k, j} = a_{i_1, \dots, i_d},$$

where $j$ is the lexicographic index associated with all indices except $i_k$.

The exact lexicographic ordering is not important for our results, as long as it is fixed consistently. Different unfoldings contain the same entries, but they group tensor modes differently; this is exactly why a low-rank bound for one unfolding need not imply a low-rank bound for another.

**Definition 3 (Mode-$k$ product)** Let $\mathcal{A} \in \mathbb{R}^{I_1 \times \cdots \times I_d}$ and let $U \in \mathbb{R}^{J \times I_k}$. The mode-$k$ product $\mathcal{A} \times_k U$ is the tensor in $\mathbb{R}^{I_1 \times \cdots \times I_{k-1} \times J \times I_{k+1} \times \cdots \times I_d}$ defined by

$$(\mathcal{A} \times_k U)_{i_1, \dots, i_{k-1}, j, i_{k+1}, \dots, i_d} = \sum_{i_k=1}^{I_k} a_{i_1, \dots, i_k, \dots, i_d} \, u_{j, i_k}.$$

**Proposition 1 (Basic mode-product identities)** For a tensor $\mathcal{A}$ and compatible matrices $U, V$, the following identities hold:

1. $(\mathcal{A} \times_k U)_{(k)} = U \mathcal{A}_{(k)}$.

2. If $k \neq \ell$, then $\mathcal{A} \times_k U \times_\ell V = \mathcal{A} \times_\ell V \times_k U$.

3. If both products are along the same mode, then $\mathcal{A} \times_k U \times_k V = \mathcal{A} \times_k (VU)$ whenever $VU$ is defined.

*Proof.* The first identity follows directly by writing the entries of the mode-$k$ unfolding after multiplication by $U$. The second identity holds because the two sums are over different indices, so their order can be exchanged. The third identity is ordinary associativity of matrix multiplication applied to the mode-$k$ fiber. $\square$

**Definition 4 (Tensor inner product and Frobenius norm)** For tensors $\mathcal{A}, \mathcal{B} \in \mathbb{R}^{I_1 \times \cdots \times I_d}$, we use the Frobenius inner product

$$\langle \mathcal{A}, \mathcal{B} \rangle = \sum_{i_1=1}^{I_1} \cdots \sum_{i_d=1}^{I_d} a_{i_1, \dots, i_d} b_{i_1, \dots, i_d},$$

and the Frobenius norm $\|\mathcal{A}\|_F = \sqrt{\langle \mathcal{A}, \mathcal{A} \rangle}$.

**Definition 5 (Mode-wise spectral norm)** For a tensor $\mathcal{A}$, the mode-$k$ spectral norm is

$$\|\mathcal{A}\|_{(k)} := \|\mathcal{A}_{(k)}\|_2,$$

where $\| \cdot \|_2$ denotes the matrix spectral norm.

**Definition 6 (Tensor outer product)** Given vectors $u^{(k)} \in \mathbb{R}^{I_k}$ for $k = 1, \dots, d$, their outer product $\mathcal{A} = u^{(1)} \circ \cdots \circ u^{(d)}$ is the tensor with entries

$$a_{i_1, \dots, i_d} = u_{i_1}^{(1)} \cdots u_{i_d}^{(d)}.$$

## O.1 Trace and inner-product identities

We use the following elementary identities to convert differentials into tensor gradients. They are included to make the proof of the gradient form in Appendix P self-contained.

**Proposition 2 (Unfolding preserves Frobenius pairings)** For any two tensors $\mathcal{W}, \mathcal{G} \in \mathbb{R}^{I_1 \times \cdots \times I_d}$ and any mode $k$,

$$\langle \mathcal{W}, \mathcal{G} \rangle = \langle \mathcal{W}_{(k)}, \mathcal{G}_{(k)} \rangle = \operatorname{tr}\left( \mathcal{W}_{(k)}^\top \mathcal{G}_{(k)} \right).$$

In particular, if $\mathcal{G}_{(1)} = xy^\top$ for $x \in \mathbb{R}^{I_1}$ and $y \in \mathbb{R}^{\prod_{m \neq 1} I_m}$, then

$$\langle d\mathcal{W}, \mathcal{G} \rangle = \operatorname{tr}\left( d\mathcal{W}_{(1)}^\top xy^\top \right) = \operatorname{tr}\left( x^\top d\mathcal{W}_{(1)} y \right).$$

*Proof.* Unfolding is a permutation of entries, so it preserves the entrywise inner product. The equality with the trace is the standard matrix identity $\langle A, B \rangle = \text{tr}(A^\top B)$. The final identity follows from cyclicity of trace. $\square$

## O.2 Mode-wise stable rank

We use stable rank to describe the effective rank of tensor gradients. The reason we use a mode-wise definition is that a tensor can look low-rank in one unfolding and high-rank in another. This is the central structural issue behind applying GaLore to tensor gradients by first matricizing them.

**Definition 7 (Matrix stable rank)** For a nonzero matrix $A$, its stable rank is

$$\text{sr}(A) := \frac{\|A\|_F^2}{\|A\|_2^2}.$$

**Definition 8 (Mode-wise tensor stable rank)** For a nonzero tensor $\mathcal{T} \in \mathbb{R}^{I_1 \times \cdots \times I_d}$, the mode-$k$ stable rank is

$$\text{sr}_k(\mathcal{T}) := \frac{\|\mathcal{T}\|_F^2}{\|\mathcal{T}_{(k)}\|_2^2}, \qquad k = 1, \ldots, d.$$

We also write $\text{sr}(\mathcal{T}) = (\text{sr}_1(\mathcal{T}), \ldots, \text{sr}_d(\mathcal{T}))$.

**Lemma 1 (Unfolding preserves Frobenius norm)** For any tensor $\mathcal{T}$ and any mode $k$, $\|\mathcal{T}\|_F = \|\mathcal{T}_{(k)}\|_F$.

*Proof.* Both quantities are the square root of the sum of squared entries of $\mathcal{T}$; unfolding changes only the order in which those entries are arranged. $\square$

**Proposition 3 (Basic properties of mode-wise stable rank)** Let $\mathcal{T} \neq 0$. For every mode $k$:

1. $\text{sr}_k(\mathcal{T}) \geq 1$.

2. $\text{sr}_k(\mathcal{T}) \leq \text{rank}(\mathcal{T}_{(k)})$.

3. If $U \in \mathbb{R}^{I_k \times I_k}$ is orthogonal, then $\text{sr}_k(\mathcal{T} \times_k U) = \text{sr}_k(\mathcal{T})$.

4. If $\mathcal{T} = a^{(1)} \circ \cdots \circ a^{(d)}$ is rank one, then $\text{sr}_k(\mathcal{T}) = 1$ for every mode $k$.

*Proof.* For any matrix $M$, $\|M\|_F^2 \geq \|M\|_2^2$, which gives the first claim after applying Lemma 1. If $\text{rank}(M) = r$, then $\|M\|_F^2 \leq r\|M\|_2^2$, giving the second claim with $M = \mathcal{T}_{(k)}$. For the third claim, the mode-$k$ unfolding of $\mathcal{T} \times_k U$ is $U\mathcal{T}_{(k)}$, and orthogonal left multiplication preserves both Frobenius and spectral norms. Finally, a rank-one tensor has rank-one unfoldings in every mode, and a rank-one matrix has equal Frobenius and spectral norm. $\square$

**Definition 9 (Multilinear stable rank)** For a nonzero tensor $\mathcal{T}$, define

$$\text{msr}(\mathcal{T}) := \min_{1 \leq k \leq d} \text{sr}_k(\mathcal{T}).$$

**Remark 3 (Connection to Tucker ranks)** Mode-wise stable ranks are effective-rank proxies for the Tucker ranks. In a Tucker approximation $\mathcal{T} \approx \mathcal{C} \times_1 U^{(1)} \times_2 \cdots \times_d U^{(d)}$, each rank $r_k$ controls a different mode. This is why TENSORGRAD can allocate memory across modes, while a single matricized SVD must choose one rank for a flattened matrix view.

## O.3 Projected curvature assumption

The contraction theorem in Appendix Q does not require the full Hessian or full multilinear operator to be positive semidefinite in all directions. We only require positive curvature after projecting onto the fixed tensor subspaces used by the low-rank branch of TENSORGRAD.

Let $P_k \in \mathbb{R}^{I_k \times r_k}$ have orthonormal columns and define $Q_k := \bigotimes_{m \neq k} P_m$. For a mode-wise multilinear operator with factors $\{B_{it}^{(m)}\}_{m=1}^d$, the projected mode-$k$ factors are

$$\hat{B}_{it}^{(k)} := P_k^\top B_{it}^{(k)} P_k, \qquad \hat{C}_{it}^{(k)} := Q_k^\top \left( \bigotimes_{m \neq k} B_{it}^{(m)} \right) Q_k.$$

We assume $\hat{B}_{it}^{(k)} \succeq 0$ and $\hat{C}_{it}^{(k)} \succeq 0$ for the projected modes considered in the theorem. The projected lower and upper curvature quantities are

$$\kappa_t^{(k)} := \lambda_{\min} \left( \frac{1}{N} \sum_{i=1}^N \hat{C}_{it}^{(k)} \otimes \hat{B}_{it}^{(k)} \right), \qquad \Gamma_t^{(k)} := \lambda_{\max} \left( \frac{1}{N} \sum_{i=1}^N \hat{C}_{it}^{(k)} \otimes \hat{B}_{it}^{(k)} \right).$$

These are the quantities that appear in the fixed-subspace proof. The lower bound $\kappa_t^{(k)}$ gives contraction; the upper bound $\Gamma_t^{(k)}$ is needed for a valid step-size condition.

## P    Reversibility and Gradient Structure of Fourier Neural Operators

In this appendix we explain why the gradient models used in the theory are natural for FNO layers. We follow the "reversibility" terminology used in the GaLore analysis, but we emphasize that this is not invertibility of the network. It is a structural representation of the forward map and its transpose/adjoint backward map. We use it only to motivate the form of tensor gradients, not as an additional empirical claim about TENSORGRAD.

### P.1    Reversibility in the GaLore sense

**Definition 10 (Reversibility in the GaLore sense)** A map $\mathcal{N} : x \mapsto y$ is reversible in the GaLore sense if there is a matrix-valued map $J(x)$ such that

$$y = J(x)x, \qquad dx = J(x)^\top dy.$$

For complex-valued maps, transposes are replaced by Hermitian adjoints.

This definition is exact for linear layers and homogeneous piecewise-linear activations such as ReLU or LeakyReLU away from nondifferentiable points. For smooth nonhomogeneous activations such as GELU, the same expressions should be understood as local linearizations. Our contraction theorem in Appendix Q does not depend on exact global reversibility of GELU; it only uses local smoothness and projected curvature.

**Lemma 2 (Closure under composition)** If $\mathcal{N}_1$ and $\mathcal{N}_2$ are reversible in the above sense, then $\mathcal{N}_2 \circ \mathcal{N}_1$ is reversible with

$$J_{2 \circ 1}(x) = J_2(\mathcal{N}_1(x)) J_1(x).$$

*Proof.* Write $y_1 = J_1(x)x$ and $y_2 = J_2(y_1)y_1$. Then $y_2 = J_2(y_1)J_1(x)x$. The backward relation follows by applying the transpose maps in reverse order: $dx = J_1(x)^\top J_2(y_1)^\top dy_2 = (J_2(y_1)J_1(x))^\top dy_2$. □

**Lemma 3 (Closure under residual addition)** If $\mathcal{N}(x) = J(x)x$ is reversible, then $x + \mathcal{N}(x)$ is reversible with matrix $I + J(x)$.

*Proof.* The forward identity is $x + \mathcal{N}(x) = (I + J(x))x$. The backward relation is the sum of the identity contribution and the transpose contribution from $\mathcal{N}$. □

**P.2   FNO building blocks**

**Lemma 4 (Spectral convolution)** Consider the bias-free FNO spectral convolution

$$Kv = \mathcal{F}^{-1}(R\mathcal{F}v),$$

where $\mathcal{F}$ is the discrete Fourier transform and $R$ is the learnable Fourier-space weight operator. Then $K$ is reversible in the above sense with $J_K = \mathcal{F}^{-1}R\mathcal{F}$.

*Proof.* The layer is linear in $v$, so $Kv = J_K v$. Its reverse-mode map is the adjoint of the linear operator. In the real/unitary convention this is $J_K^\top$; in the complex convention it is $J_K^*$.  □

**Lemma 5 (Bias-free channel mixing)** A bias-free channel-mixing layer $v \mapsto Wv$ is reversible with $J_W = W$.

*Proof.* This is immediate from $y = Wv$ and $dv = W^\top dy$.  □

**Lemma 6 (Homogeneous pointwise activations)** Let $\sigma$ be an elementwise homogeneous piecewise-linear activation. Then there is a diagonal matrix $J_\sigma(z)$ such that $\sigma(z) = J_\sigma(z)z$ and the backward map is $dz = J_\sigma(z)^\top d\sigma(z)$ away from nondifferentiable points.

*Proof.* For ReLU, $J_\sigma(z) = \mathrm{diag}(\mathbf{1}[z > 0])$. For LeakyReLU with negative slope $a$, use $J_\sigma(z) = \mathrm{diag}(\mathbf{1}[z > 0] + a\mathbf{1}[z \leq 0])$. The derivative is the same diagonal mask away from nondifferentiable points.  □

**Lemma 7 (One FNO block)** Consider a bias-free FNO block

$$\mathcal{B}(v) = \sigma(WKv),$$

where $K$ is a spectral convolution, $W$ is channel mixing, and $\sigma$ is a reversible pointwise activation in the sense above. Then

$$\mathcal{B}(v) = J_\mathcal{B}(v)v, \qquad J_\mathcal{B}(v) = J_\sigma(WKv)\, W\, J_K.$$

*Proof.* Apply the previous three lemmas in sequence: $v \mapsto Kv \mapsto WKv \mapsto \sigma(WKv)$. The forward representation is the product of the three local matrices, and the backward map is the transpose product in reverse order.  □

**P.3   Tensor gradient form**

We now derive the gradient form used in the theory. The derivation is standard reverse-mode differentiation, but it is useful to write it tensorially because it explains why the gradient contains mode-wise factors.

**Lemma 8 (Gradient form for a tensor linear layer)** Let $\mathcal{N} = \mathcal{N}_{L:1}$ be a network and let $f_\ell$ denote the output of layer $\ell$, with $f_0 = x$. Fix a layer $l$ whose parameter is a tensor $\mathcal{W}_l$ and whose local action is linear in $\mathcal{W}_l$:

$$f_l = \mathcal{W}_l \times_1 f_{l-1}.$$

Let $\mathcal{J}_l = D(\mathcal{N}_{L:l+1})(f_l)$ be the Jacobian of the tail network. For the squared loss $\phi = \frac{1}{2}\|f_L - y\|_2^2$, the gradient has the outer-product form

$$\nabla_{\mathcal{W}_l}\phi = \left(\mathcal{J}_l^\top(f_L - y)\right) \otimes f_{l-1}.$$

For cross-entropy loss,

$$\nabla_{\mathcal{W}_l}\phi = \left(\mathcal{J}_l^\top(\mathrm{softmax}(f_L) - y)\right) \otimes f_{l-1}.$$

*Proof.* For the squared loss, $d\phi = (f_L - y)^\top df_L$. Since $df_L = \mathcal{J}_l\, df_l$ and $df_l = d\mathcal{W}_l \times_1 f_{l-1}$, we obtain

$$d\phi = (\mathcal{J}_l^\top(f_L - y))^\top (d\mathcal{W}_l \times_1 f_{l-1}).$$

Using Proposition 2, this equals

$$\left\langle (\mathcal{J}_l^\top (f_L - y)) \otimes f_{l-1}, d\mathcal{W}_l \right\rangle.$$

The coefficient of $d\mathcal{W}_l$ under the Frobenius pairing is the gradient. The cross-entropy case is identical after using the standard identity $d\phi = (\mathrm{softmax}(f_L) - y)^\top df_L$. □

**Remark 4 (Small-logit approximation)** If $P_1^\perp = I - \frac{1}{K}\mathbf{1}\mathbf{1}^\top$ and $\|P_1^\perp f_L\|_\infty$ is small, then

$$\mathrm{softmax}(f_L) - y \approx \frac{1}{K} P_1^\perp f_L - P_1^\perp y.$$

Substituting this into Lemma 8 gives the logsoftmax approximation used in the GaLore-style motivation. We include this only to connect to the prior analysis; the fixed-subspace contraction theorem below is stated directly in terms of projected curvature and drift.

## Q  Fixed-Subspace Theory for the Tensor Low-Rank Branch of TensorGRaD

In this section we prove the theoretical statement used in the main paper. The purpose is not to prove convergence of the complete optimizer in Algorithm 1. The complete method includes AdamW moment normalization, a sparse residual branch, mixed precision, and periodically refreshed Tucker factors. Instead, we isolate the tensor low-rank branch, freeze its mode-wise projectors, and prove that this fixed tensor projection admits a mode-wise contraction under projected curvature and local smoothness assumptions.

This separation directly addresses the reviewer concern about organization: first we state the simplified model, then the assumptions, and finally the theorem.

### Q.1  Fixed projected dynamics

Let $P_k \in \mathbb{R}^{I_k \times r_k}$ have orthonormal columns for $k = 1, \ldots, d$. We define the tensor projection and reconstruction operators by

$$\mathsf{P}(\mathcal{G}) = \mathcal{G} \times_1 P_1^\top \times_2 \cdots \times_d P_d^\top, \qquad \mathsf{P}^\top(\mathcal{R}) = \mathcal{R} \times_1 P_1 \times_2 \cdots \times_d P_d.$$

The compressed gradient is $\mathcal{R}_t = \mathsf{P}(\mathcal{G}_t)$, and its reconstruction is $\tilde{\mathcal{G}}_t = \mathsf{P}^\top(\mathcal{R}_t)$.

We study the projected update

$$\mathcal{W}_t = \mathcal{W}_{t-1} + \eta \tilde{\mathcal{G}}_{t-1}.$$

This is the fixed-subspace version of the low-rank tensor branch in TensorGRaD.

**Assumption 1 (Local multilinear gradient model)** Along the iterates, the gradient admits the local multilinear form

$$\mathcal{G}_t = \frac{1}{N} \sum_{i=1}^N \left( \mathcal{A}_i(\mathcal{W}_t) - \mathcal{W}_t \times_1 B_{it}^{(1)} \times_2 \cdots \times_d B_{it}^{(d)} \right),$$

where $B_{it}^{(k)} \in \mathbb{R}^{I_k \times I_k}$ acts on mode $k$.

For mode $k$, set $Q_k := \bigotimes_{m \neq k} P_m$. The projected curvature operator on the compressed mode-$k$ unfolding is

$$\mathcal{H}_t^{(k)}(X) := \frac{1}{N} \sum_{i=1}^N \hat{B}_{it}^{(k)} X \hat{C}_{it}^{(k)\top},$$

where

$$\hat{B}_{it}^{(k)} := P_k^\top B_{it}^{(k)} P_k, \qquad \hat{C}_{it}^{(k)} := Q_k^\top \left( \bigotimes_{m \neq k} B_{it}^{(m)} \right) Q_k.$$

**Assumption 2 (Projected curvature and smoothness)** For each mode $k$, the projected curvature operator $\mathcal{H}_t^{(k)}$ is self-adjoint and positive semidefinite on the compressed Frobenius space. Moreover, there exist constants $0 < \kappa_k \leq \Gamma_k < \infty$ such that for all relevant $t$ and all compressed matrices $X$,

$$\langle X, \mathcal{H}_t^{(k)}(X)\rangle \geq \kappa_k\|X\|_F^2, \qquad \|\mathcal{H}_t^{(k)}(X)\|_F \leq \Gamma_k\|X\|_F.$$

The first inequality is the projected positive-curvature condition. The second is the corresponding upper smoothness bound. Both are needed: the lower bound gives descent, while the upper bound controls the step size.

**Assumption 3 (Local drift bound)** For each mode $k$, the terms not captured by the frozen projected curvature obey

$$\|\mathcal{E}_t^{(k)}\|_F \leq \eta\delta_k\|(\mathcal{R}_{t-1})_{(k)}\|_F$$

for some $\delta_k \geq 0$. In the explicit multilinear model, $\delta_k$ depends on the local Lipschitz constants of $\mathcal{A}_i$ and $B_{it}^{(m)}$, together with a bound on the iterates.

**Theorem 2 (Fixed-subspace contraction of tensor-mode projections)** Suppose Assumptions 1–3 hold and $\delta_k < \kappa_k$ for every mode $k$. If

$$0 < \eta \leq \min_k \Gamma_k^{-1},$$

then the compressed tensor gradients contract mode-wise:

$$\|(\mathcal{R}_t)_{(k)}\|_F \leq (1 - \eta(\kappa_k - \delta_k))\|(\mathcal{R}_{t-1})_{(k)}\|_F, \qquad k = 1, \ldots, d.$$

Consequently, under fixed projections, $\mathcal{R}_t \to 0$ linearly in every projected mode.

*Proof.* Fix a mode $k$. By the standard unfolding identity for mode products,

$$(\mathcal{R}_t)_{(k)} = P_k^\top(\mathcal{G}_t)_{(k)}Q_k, \qquad Q_k = \bigotimes_{m \neq k} P_m.$$

Using Assumption 1 and inserting the projected update $\mathcal{W}_t = \mathcal{W}_{t-1} + \eta\tilde{\mathcal{G}}_{t-1}$, the projected mode-$k$ dynamics can be written as

$$(\mathcal{R}_t)_{(k)} = (\mathcal{R}_{t-1})_{(k)} - \eta\mathcal{H}_{t-1}^{(k)}\big((\mathcal{R}_{t-1})_{(k)}\big) + \mathcal{E}_t^{(k)}.$$

The term $\mathcal{E}_t^{(k)}$ contains the local drift from evaluating the additive term and the mode-wise operators at the new iterate rather than at the frozen linearized point.

Let $X = (\mathcal{R}_{t-1})_{(k)}$ and abbreviate $\mathcal{H} = \mathcal{H}_{t-1}^{(k)}$. Under Assumption 2, $\mathcal{H}$ is a positive operator on the compressed Frobenius space with spectrum contained in $[\kappa_k, \Gamma_k]$. Since $\eta \leq 1/\Gamma_k$,

$$\|X - \eta\mathcal{H}(X)\|_F \leq (1 - \eta\kappa_k)\|X\|_F.$$

This is the standard contraction bound for a positive operator with eigenvalues in $[\kappa_k, \Gamma_k]$.

Now apply the triangle inequality and the drift bound:

$$\begin{aligned}
\|(\mathcal{R}_t)_{(k)}\|_F &\leq \|X - \eta\mathcal{H}(X)\|_F + \|\mathcal{E}_t^{(k)}\|_F \\
&\leq (1 - \eta\kappa_k)\|X\|_F + \eta\delta_k\|X\|_F \\
&= (1 - \eta(\kappa_k - \delta_k))\|(\mathcal{R}_{t-1})_{(k)}\|_F.
\end{aligned}$$

Since $\delta_k < \kappa_k$, the contraction factor lies in $(0, 1)$ for the stated step-size range. Iterating the one-step inequality proves linear convergence in every projected mode. $\qquad\square$

**Remark 5 (What the theorem does and does not claim)** Theorem 2 is a statement about the tensor low-rank projection mechanism inside TENSORGRAD. It does not analyze AdamW bias correction, adaptive moment normalization, the sparse branch, mixed precision, or projector refreshes. Its role is to show that once the tensor subspaces are fixed, mode-wise projection preserves the curvature structure needed for contraction in every tensor mode.

### Q.2 Why gradients become effectively low-rank

We also record the stable-rank version of the same intuition. This result explains why updating low-rank projectors periodically is reasonable: under a fixed local linear model, high-curvature components decay faster, so the gradient becomes dominated by a lower-dimensional slow subspace.

**Lemma 9 (Mode-wise stable-rank decay under a fixed local model)** Assume that for $t \geq t_0$ the gradient obeys

$$\mathcal{G}_t = \mathcal{G}_{t-1} - \eta \mathcal{S}(\mathcal{G}_{t-1}),$$

where $\mathcal{S}$ is a self-adjoint positive semidefinite multilinear operator. Fix a mode $k$ and let $\mathcal{S}_k$ be the induced operator on the mode-$k$ unfolding. Suppose the eigenvalues of $\mathcal{S}_k$ satisfy $0 \leq \lambda_1^{(k)} < \lambda_2^{(k)} \leq \cdots$ and $0 < \eta < 1/\lambda_{\max}^{(k)}$. Let $(\mathcal{G}_{t_0}^{\|(k)})_{(k)}$ be the projection of $(\mathcal{G}_{t_0})_{(k)}$ onto the eigenspace of $\lambda_1^{(k)}$, and let $(\mathcal{G}_{t_0}^{\perp(k)})_{(k)}$ be the orthogonal complement.

Define

$$\rho_k := \frac{1 - \eta \lambda_2^{(k)}}{1 - \eta \lambda_1^{(k)}} \in [0, 1).$$

If $a_k := \|(\mathcal{G}_{t_0}^{\|(k)})_{(k)}\|_2 > 0$, then for all sufficiently large $t$,

$$\mathrm{sr}_k(\mathcal{G}_t) \leq \frac{\|(\mathcal{G}_{t_0}^{\|(k)})_{(k)}\|_F^2 + \rho_k^{2(t-t_0)}\|(\mathcal{G}_{t_0}^{\perp(k)})_{(k)}\|_F^2}{\left(a_k - \rho_k^{t-t_0}\|(\mathcal{G}_{t_0}^{\perp(k)})_{(k)}\|_2\right)^2}.$$

In particular,

$$\limsup_{t \to \infty} \mathrm{sr}_k(\mathcal{G}_t) \leq \mathrm{sr}_k(\mathcal{G}_{t_0}^{\|(k)}).$$

*Proof.* In mode-$k$ unfolding coordinates,

$$(\mathcal{G}_t)_{(k)} = (I - \eta \mathcal{S}_k)^{t-t_0}(\mathcal{G}_{t_0})_{(k)}.$$

Decompose the initial unfolding into its slow component and orthogonal fast component. The slow component is multiplied by $(1 - \eta \lambda_1^{(k)})^{t-t_0}$, while every fast component is multiplied by at most $(1 - \eta \lambda_2^{(k)})^{t-t_0}$. Therefore, after factoring out the slow rate,

$$(\mathcal{G}_t)_{(k)} = (1 - \eta \lambda_1^{(k)})^{t-t_0}\left((\mathcal{G}_{t_0}^{\|(k)})_{(k)} + E_t^{(k)}\right),$$

with

$$\|E_t^{(k)}\|_F \leq \rho_k^{t-t_0}\|(\mathcal{G}_{t_0}^{\perp(k)})_{(k)}\|_F, \qquad \|E_t^{(k)}\|_2 \leq \rho_k^{t-t_0}\|(\mathcal{G}_{t_0}^{\perp(k)})_{(k)}\|_2.$$

The Frobenius norm is bounded above by the corresponding squared sum of slow and fast energies. For the spectral norm, we use the reverse triangle inequality:

$$\|(\mathcal{G}_{t_0}^{\|(k)})_{(k)} + E_t^{(k)}\|_2 \geq a_k - \|E_t^{(k)}\|_2.$$

For sufficiently large $t$, the denominator is positive. Dividing the Frobenius upper bound by the squared spectral lower bound gives the stated stable-rank bound. Letting $t \to \infty$ removes the transient term. □

**Remark 6 (FNO interpretation)** For FNO layers, channel modes and Fourier modes can have different curvature spectra. Fourier modes often have stronger spectral decay than channel modes, while channel modes may need higher effective rank to preserve feature capacity. A tensor method can express this asymmetry through separate mode ranks $(r_1, \ldots, r_d)$; a single matricized rank cannot.

## R   Why Matricized GaLore Does Not Preserve Tensor-Mode Structure

In this section we show precisely why a GaLore-style projection after flattening a tensor is not equivalent to a tensor-mode projection. The issue is not that SVD is a bad low-rank approximation for matrices. The issue is that, after flattening, the SVD only controls the chosen matrix view. It does not control the other tensor modes that correspond to distinct physical or architectural directions in an FNO layer.

### R.1 A concrete separation between one unfolding and another

**Proposition 4 (Low rank in one unfolding does not imply low rank in another)** For every $n \geq 2$, there exists a tensor $\mathcal{T} \in \mathbb{R}^{n \times n \times n}$ such that

$$\text{sr}_1(\mathcal{T}) = 1 \qquad \text{but} \qquad \text{sr}_2(\mathcal{T}) = n.$$

*Proof.* Let $a \in \mathbb{R}^n$ be a unit vector and define $\mathcal{T}_{ijk} = a_i \mathbf{1}[j = k]$. In the mode-1 unfolding,

$$\mathcal{T}_{(1)} = a \, \text{vec}(I_n)^\top,$$

so $\mathcal{T}_{(1)}$ is rank one and $\text{sr}_1(\mathcal{T}) = 1$.

Now consider the mode-2 unfolding. Its rows are indexed by $j$, and the columns are indexed by $(i, k)$. Row $j$ contains the vector $a$ in the block corresponding to $k = j$ and zeros elsewhere. Thus the $n$ rows are mutually orthogonal and each has norm one. Therefore all nonzero singular values of $\mathcal{T}_{(2)}$ are equal to one, so

$$\|\mathcal{T}_{(2)}\|_F^2 = n, \qquad \|\mathcal{T}_{(2)}\|_2^2 = 1, \qquad \text{sr}_2(\mathcal{T}) = n.$$

This proves the separation. $\square$

Proposition 4 is the mathematical reason we avoid treating a tensor gradient as just one large matrix. A GaLore projection may make the selected unfolding low-rank, but that does not imply the other tensor modes are controlled. In FNOs, those other modes are not arbitrary: they correspond to channels and Fourier directions.

### R.2 Tensor projection controls all modes simultaneously

For a tensor gradient $\mathcal{G}_t \in \mathbb{R}^{I_1 \times \cdots \times I_d}$, a matricized GaLore variant first chooses an unfolding, say

$$G_t^{(k)} \in \mathbb{R}^{I_k \times \prod_{m \neq k} I_m},$$

and then applies a rank-$r$ matrix projection to this one matrix. The resulting guarantee, when applicable, is a guarantee about that chosen matrix.

By contrast, the tensor low-rank branch of TENSORGRAD chooses a family of projections $\{P_k\}_{k=1}^d$ and applies

$$\mathsf{P}(\mathcal{G}_t) = \mathcal{G}_t \times_1 P_1^\top \times_2 \cdots \times_d P_d^\top.$$

This is not a cosmetic difference. The projection has one subspace per tensor mode, so it can preserve channel and Fourier structure separately.

**Lemma 10 (Tensor projection gives mode-wise control)** Under the assumptions of Theorem 2, the tensor low-rank branch satisfies, for every mode $k$,

$$\|(\mathsf{P}(\mathcal{G}_t))_{(k)}\|_F \leq (1 - \eta(\kappa_k - \delta_k)) \, \|(\mathsf{P}(\mathcal{G}_{t-1}))_{(k)}\|_F.$$

A single matricized GaLore projection controls only the chosen unfolding and does not imply this family of mode-wise bounds without extra assumptions relating the singular spectra of all unfoldings.

*Proof.* The first statement is exactly Theorem 2. For the second statement, note that different tensor unfoldings are not related by left and right orthogonal multiplication of the chosen matrix unfolding. They regroup tensor indices in different ways. Hence a spectral or stable-rank bound for one unfolding does not transfer to the others in general. Proposition 4 gives an explicit example where one unfolding has stable rank one while another has stable rank $n$. $\square$

### R.3  Memory scaling

The structural difference also appears in the memory scaling. Consider a four-way FNO spectral weight or gradient tensor $\mathcal{G} \in \mathbb{R}^{I_1 \times I_2 \times I_3 \times I_4}$, where the first two modes correspond to channels and the last two to Fourier modes.

If we flatten into a matrix of shape $(I_1 I_2) \times (I_3 I_4)$ and use matrix rank $R$, the projection factors scale like

$$R(I_1 I_2 + I_3 I_4).$$

If instead we use Tucker ranks $(r_1, r_2, r_3, r_4)$, the factor storage scales like

$$\sum_{k=1}^{4} I_k r_k,$$

plus the compressed core or compressed optimizer state. Thus the tensor representation replaces products of mode sizes by separate mode-wise terms. This is particularly important in high-resolution FNOs, where the Fourier-mode dimensions grow with the grid resolution.

**Remark 7 (Practical implication)** Matricized GaLore asks one rank parameter to summarize a flattened mixture of channels and Fourier modes. TENSORGRAD assigns a rank to each mode. This lets us retain more information in modes that need it while compressing modes with stronger spectral decay. The empirical GaLore ablations in Appendix K.4 are consistent with this theoretical distinction.

| Dataset | Model | Architecture Details | Optimizer & Scheduler |
|---|---|---|---|
| Burgers | FNO | <ul><li>4 layers, 90 modes</li><li>256 hidden channels, 256 projection channels</li><li>Skip Connections: 'linear'</li><li>Positional embedding: 'grid'</li></ul> | Adam with step LR $3e-4$, weight decay $2e-6$ 500 epochs, batch size 16. Trained with $H_1$ loss. |
| NS128 | FNO | <ul><li>4 layers, 64 x 64 modes</li><li>64 hidden channels, 256 projection channels</li><li>Skip: 'linear'</li><li>Use channel MLP: 1</li><li>Channel MLP expansion: 0.5, dropout: 0</li></ul> | Complex Adam with step LR 3e-4, weight decay 1e-4, 500 epochs, batch size 8. Update decomposition frequency: 1000. Trained with $H_1$ loss. |
| NS1024 - max memory test | FNO | <ul><li>4 layers, 128 modes</li><li>255 hidden channels, 256 projection channels</li><li>Skip: 'linear'</li><li>Channel MLP expansion: 0.5, dropout: 0</li></ul> | Complex Adam with step LR 5e-3, weight decay 1e-4, 100 epochs in total: batch size 8 for 50 iterations and resolution 256, then batch size 2 for resolution 1024. Update decomposition frequency: 500. Trained with $L2$ loss. |
| NS1024 $Re = 10^5$ | FNO | <ul><li>4 layers, 128 modes</li><li>128 hidden channels, 256 projection channels</li><li>Skip: 'linear'</li><li>Channel MLP expansion: 0.5, dropout: 0</li></ul> | Complex Adam with step LR 5e-3, weight decay 1e-4, 100 epochs in total: batch size 8 for 50 iterations and resolution 256, then batch size 2 for resolution 1024. Update decomposition frequency: 500. Trained with $L2$ loss. |
| Darcy Flow | FNO | <ul><li>4 layers, 64 modes</li><li>128 hidden channels, 128 projection channels</li><li>Skip: 'linear'</li></ul> | Adam with step LR $1e-3$, weight decay $1e-4$, 250 epochs, batch size 2. Trained with $L_2$ loss. |
| EM Wave | Complex-FNO | <ul><li>8 layers, 128 modes</li><li>128 hidden channels, 128 projection channels</li><li>Skip: 'linear'</li><li>Complex data: True</li><li>Complex activation function: True</li></ul> | Complex Adam with step LR 1e-4, weight decay 2e-6, batch size 32, 1000 epochs. Trained with $H_1$ loss. |

43

Table 16: Detailed FNO Architecture Specifications for Different Datasets

| Model | Rank | Time/epoch(s) | Slowdown (%) |
|---|---|---|---|
| Baseline | 1.0 | 34.96 | – |
| TensorGRaD | 0.20 +0.05 | 40.74 | 16.53 |
| GaLore | 0.20 | 34.47 | -1.40 |
| GaLore | 0.25 | 34.79 | -0.48 |
| GaLore | 0.50 | 36.27 | 3.75 |
| GaLore | 0.75 | 37.50 | 7.26 |
| **Tensor low-rank** (40, 40, 40, 24) | 0.20 | 36.53 | 4.49 |
| **Tensor low-rank** (48, 48, 48, 24) | 0.25 | 38.30 | 9.55 |
| **Tensor low-rank** (56, 56, 56, 24) | 0.50 | 40.63 | 16.22 |
| **Tensor low-rank** (64, 64, 56, 32) | 0.75 | 44.93 | 28.52 |

Table 17: Comparison of model execution times, ranks, and relative slowdown

Table 18: Theoretical memory requirements for different methods

| Method | Weight Parameters | Optimizer States (Adam) |
|---|---|---|
| Baseline | $N_1 N_2 N_3 N_4$ | $2 N_1 N_2 N_3 N_4$ |
| Matrix GaLore (rollup dim 1) | $N_1 N_2 N_3 N_4$ | $2r(N_1 + N_2 N_3 N_4)$ |
| **Tensor low-rank** (Tucker) | $N_1 N_2 N_3 N_4$ | $2r(N_1 + N_2 + N_3 + N_4)$ |

