# OpenReview forum: "TensorGRaD: Tensor Gradient Robust Decomposition for Memory-Efficient Neural Operator Training"
_TMLR — Decision pending for TMLR_

### Review · Reviewer_C4Mx · 2026-04-15

**Summary Of Contributions:**

The paper proposes TensorGRaD, a method for memory-efficient training of neural operators. It compresses tensor gradients with a low-rank and sparse decomposition, studies a mixed-precision setting, provides theory for tensor-mode projection, and reports results on several PDE benchmarks.

**Audience:**

Yes

**Audience Explanation:**

The paper studies memory-efficient training for neural operators, which is relevant to researchers working on scientific machine learning and large-scale models.

**Broader Impact Concerns:**

No major ethical concerns are identified. The work focuses on improving training efficiency for neural operators and is mainly technical in nature. Any potential impact is indirect, such as enabling larger-scale models, which is already common in the field.

**Claims And Evidence:**

Yes

**Claims Explanation:**

The main claims are supported by both theory and experiments. The paper gives an analysis for tensor-mode projection and also reports results on several PDE benchmarks, including comparisons with low-rank, sparse, and matrix-based baselines.

**Requested Changes:**

1 (Theoretical scope). Clarify the scope of the theoretical analysis more explicitly in the main paper. The current theory is stated for a simplified setting with fixed mode-wise projections, and it does not cover the full AdamW update, the sparse residual branch, or time-varying projections. Making this limitation more prominent would improve clarity.

2 (Writing). Improve the writing more carefully. (i) Fix incomplete cross-references in the appendix and proofs, such as “Sec. ??”, since they make some derivations difficult to follow. (ii) Revise the proof presentation and numbering style. Labels such as “Proof 4” are distracting and reduce readability. (iii) Improve the presentation of the theory section by separating assumptions, guarantees, and the components of the full method that are not covered by the analysis.


3 (Computation overlead). Report the computational overhead of the decomposition more explicitly in the main paper, for example by summarizing the added runtime under the chosen update frequency.

4 (Hyperparameter selection). It is helpful to provide clearer guidance on hyperparameter selection, especially Tucker rank, sparsity ratio, and projector update gap, since these choices appear important in practice.

5 (Modeling assumption). It would strengthen the paper to briefly discuss the difference between the low-rank plus sparse decomposition used here and more restrictive models that assume a tensor itself is simultaneously low-rank and sparse, such as:
  [R1] Li, X., et al. Statistical performance of convex low-rank and sparse tensor recovery. Pattern Recognition, 2019.

6 (Reproducibility). For reproducibility, I encourage the authors to provide the code during the review process, for example through an anonymous repository or supplementary material, especially since the paper states that the code will be made open-source and publicly available.

---

> ### Author Response · Authors · 2026-04-29
>
> We thank the reviewer for the constructive and encouraging feedback. We are glad that the reviewer finds the main claims supported by both the theory and experiments. In the revision, we have focused on making the manuscript clearer and more practical to use.
>
> **1.Theoretical scope (RC1).**
> We agree that the scope of the theory should be stated more explicitly. In the revised manuscript, we now make clear in the main paper theory section and theorem discussion that the analysis covers only the fixed-projection tensor low-rank component of the method. It does not cover the full AdamW dynamics, the sparse residual branch, or time-varying projections. We also clarify that the purpose of the theorem is to explain why tensor-mode-preserving projections are structurally preferable to matricization, rather than to provide a full convergence theorem for TensorGRaD as implemented.
>
> **2.Writing and proof presentation (RC2).**
> We agree and have revised the theory presentation substantially for clarity. In particular, we fixed incomplete cross-references, cleaned up theorem/appendix/figure references, and reorganized the theory to more clearly separate the simplified model, assumptions, guarantee, and the parts of the full method that are outside the analysis. We also updated the appendix presentation to better reflect this structure.
>
> **3.Computational overhead (RC3).**
> We agree that the runtime overhead should be more visible in the main paper. In the revision, we move the update-frequency ablation into the main text (Fig. 6) and summarize the runtime trade-off explicitly. The main message is that the additional decomposition-related cost is modest and amortized when the projector is updated infrequently: for example, at $T=10^3$ the slowdown is about 13% for pure low-rank compression and 20% for low-rank + unstructured sparse compression. This makes TensorGRaD a practical memory-accuracy-time trade-off, rather than a method whose runtime is dominated by decomposition.
>
> **4.Hyperparameter selection (RC4).**
> We agree that the practical defaults were not sufficiently clear. In the revision, we added a short practical hyperparameter guide in Appendix C and tied it directly to the ablations. We also moved the update-frequency ablation into the main paper (Fig. 6) and added a new fixed-budget sparse-share ablation (Fig. 5) that varies the split between low-rank and sparse components at constant total optimizer size.
> Our empirical results suggest a simple default recipe: use a total optimizer budget of about 25% of Adam, assign most of this budget to the low-rank branch, keep the sparse branch small (typically 2-5%), and choose the projector update gap in the stable range $T \in [10^2,10^3]$. The new sparse-share ablation shows that a small residual sparse component improves over pure low-rank compression at matched budget, whereas larger sparse shares increase indexing overhead and degrade performance.
>
> **5. Modeling assumption: additive low-rank + sparse vs. stricter recovery models (RC5).**
> We thank the reviewer for this very helpful suggestion. We have added this distinction explicitly. In thre related work and method sections of the revised manuscript, we now clarify that TensorGRaD uses a composite low-rank + sparse approximation of the gradient, where the two components serve complementary roles during optimization. At the modeling level, this is consistent with viewing the gradient as being approximated by a low-rank part together with a sparse residual. Algorithmically, however, our method constructs these components sequentially through a residual step rather than solving a joint convex recovery problem. We contrast this with stricter tensor recovery settings that study simultaneous low-rank and sparse structure in a statistical estimation or convex recovery framework, and we cite the suggested reference in this discussion. We also make clear that TensorGRaD is a practical optimizer compression method, not a tensor recovery formulation of that kind.
>
> **6.Reproducibility / code during review (RC6).**
> We agree that this is important. We have uploaded the code as supplementary material.

---

> > ### Comment · Reviewer_C4Mx · 2026-05-06
> >
> > Thanks to the authors' feedback. I have no further questions.

---

> > > ### Author Response · Authors · 2026-05-11
> > >
> > > Thank you for the follow-up and for the constructive review. We are glad that our response addressed your concerns.

---

### Review · Reviewer_AKZm · 2026-04-15

**Summary Of Contributions:**

This paper proposes a memory-efficient optimizer for training Fourier Neural Operators by decomposing gradient tensors as a sum of a low-rank Tucker component and an unstructured sparse component before computing Adam optimizer's states.  The method avoids the multilinear structure loss that occurs when applying DVD to tensor gradients. A mixed precision variant is also presented. Experiments focus on Navier Stokes with supporting results on Burgers, Darcy, and EM propagation. The paper includes convergence analysis under the fixed mode-wise projections and a theoretical argument for why tensor decomposition is superior.

Strengths:

1. The core observation that GaLore-style matricization of tendor gradients destroys multi-linear structure is correct and clearly illustrated.

2. The comparison between tensor low-rank and matricized GaLore, is the paper's most convincing result.

3. The paper includes comparisons across a wide range of parameters, including decomposition orderings, sparsity strategies, update frequencies, and so on.

4. Mixed-precision analysis is practically relevant.

Weaknesses:

1. The claimed robust decomposition is not truly correct since no decomposition problem is solved.

2. The primary benchmark has very small test improvements that are within noise.

3. The theory covers only fixed projections and low rank SGD, explicitly excluding AdamW updates.

4. The experimental scope is narrow in the sense that all are on FNO variants solving PDEs. No wall clock memory-constrained training comparison is given.

5. Computational overhead is underreported.

6. The sparse component's contribution is unclear.

**Audience:**

Yes

**Audience Explanation:**

Please look at strengths.

**Claims And Evidence:**

Yes

**Claims Explanation:**

Please refer to the weaknesses.

**Requested Changes:**

1. Can you clarify in what sense TensorGRaD implements "robust tensor decomposition"? The algorithm is a sequential heuristic for compression, not a convex optimization over L+S. How does the theoretical machinery of Gu et al actually enter the method?

2. Why do you claim that the proposed method "surpasses" Adam rather than claiming parity when the results don't show any statistically significant gains?

3. What is the practical sensitivity to the Tucker update frequency T? At what value of T does performance degrade significantly, and what is the associated wall-clock cost?

4. For the GNO experiment, the training error increases; is this acceptable?

5. The theory excludes AdamW, the sparse component, and time-varying projections. What does Theorem 2 actually tell us about TensorGraD as implemented? Is there any theoretical result that covers the full method?

---

> ### Author Response · Authors · 2026-04-29
>
> We thank the reviewer for the careful reading and constructive feedback. We agree that the previous draft overstated some aspects of the method and theory, and we revised the manuscript to make the scope and claims more precise
>
> **1. Robust tensor decomposition terminology(W1/Q1)**
> We agree that TensorGRaD does not solve a classical RTD recovery problem at each step, and we have toned down this terminology throughout the paper. In the revision, RTD is presented only as the modeling inspiration for the low-rank+sparse view of gradients. Our contribution is to turn this viewpoint into a practical fully online optimizer compression scheme. Unlike classical low-rank + sparse recovery methods, which solve a joint optimization problem via expensive ADMM-style procedures [1], TensorGRaD uses online low-rank tensor projections, a sparse residual branch, and warm-started factor updates inside the optimizer loop. These methods recover the current tensor, but do not naturally yield a reusable projector; in practice they would require repeated solves as gradients evolve during training. By contrast, TensorGRaD maintains a projector that is stable across steps and refreshed only periodically
>
> [1] Candès et al.. “Robust Principal Component Analysis?”
>
> **2.Claims relative to Adam(W2/Q2)**
> We agree that the main empirical message should be parity at lower memory, not superiority over Adam. We had already noted in the previous version that the occasional improvement over Adam may reflect a regularization effect from compression, and in the revision we make this interpretation more explicit. We therefore revised the abstract and results discussion accordingly. Where TensorGRaD is slightly better than Adam, we now describe this as a possible regularization effect from compression, consistent with prior work on factorized structure in deep models[2], rather than as the central claim. Our primary claim is comparable accuracy at substantially lower memory
>
> [2] Kossaifi et al.“Tensor Regression Networks”
>
> **3.Theory scope(W3/Q5)**
> We now state this limitation explicitly in the main-paper theory section and theorem discussion. The analysis covers only the fixed-projection tensor low-rank component of the method; it does not cover the sparse branch, AdamW moments, changing Tucker factors, or mixed precision. We position the theorem as explaining why tensor-mode-preserving projections are structurally preferable to matricization, not as a full convergence theorem for TensorGRaD as implemented
>
> **4.Neural operators focus(W4)**
> We clarified this motivation in the introduction and discussion. Neural operators are the main target because their gradients and optimizer states are naturally higher-order tensors, optimizer memory is a severe bottleneck at high resolution, and naive matricization destroys meaningful mode structure. We retain the TFNO and GINO results in the appendix as evidence of transferability within tensor-structured operator-learning models
>
> **5.Runtime vs update-frequency(W4/W5/Q3)**
> We moved the update-gap ablation into the main paper and now report both accuracy and runtime as a function of the projector update gap T in Fig. 6. The revised text makes the trade-off explicit: frequent updates add computation and can slightly hurt due to repeated subspace switching, while very infrequent updates make the projector stale. At T=10^3, the slowdown is about 13% for pure low-rank compression and 20% for low-rank + unstructured sparse compression. We also added a short practical hyperparameter guide at the start of the appendix: start from a 25% optimizer budget, allocate most of it to the low-rank branch, keep the sparse share small (about 2-5%), and choose T in the stable range 10^2-10^3
>
> **6.Contribution of sparse branch(W6)**
> We made this point more explicit in both the paper and experiments. The sparse branch is intended as a small residual correction that captures localized outliers and heavy-tailed residual structure that pure low-rank compression misses. To isolate this, we added a new fixed-budget ablation in Fig. 5 at 25% total optimizer memory, varying the split between low-rank and unstructured sparse components. It shows that a small sparse residual (about 2–5%) improves over pure low-rank compression, while allocating too much budget to sparsity hurts performance.
>
> **7.GINO/3D result (Q4)**
> We agree that this experiment should be interpreted carefully. Our goal is not to claim that TensorGRaD optimizes GINO better than Adam, but to show that the compression idea transfers beyond 2D FNO. In the revised manuscript, we clarify this explicitly. On ShapeNet Car with 3D GINO, TensorGRaD remains close to Adam while reducing optimizer memory by 75% (train error 2.94%$\rightarrow$ 3.37%, test error 8.53%$\rightarrow$ 8.74%). We view this as evidence of applicability beyond the main 2D FNO setting, not as a claim of superiority. This gap is also small relative to the spread across model variants in the original GINO paper

---

### Review · Reviewer_3V4S · 2026-04-21

**Summary Of Contributions:**

The paper proposes TensorGRaD, a memory-efficient optimizer for neural operator training. The method decomposes tensor gradients into a low-rank tensor and a sparse tensor, which stores optimizer states in compressed form. Experiments on several PDE tasks show memory savings with competitive accuracy.

**Audience:**

Yes

**Audience Explanation:**

This paper addresses optimizer memory during training of high-resolution neural operators, which is an important issue. The proposed low-rank and sparse gradient compression is also relevant to broader discussion on efficient training.

**Broader Impact Concerns:**

No.

**Claims And Evidence:**

No

**Claims Explanation:**

1. Theory does not fully match the algorithm:
The theoretic section mainly analyzes the fixed low-rank projection case. It does not cover important parts of the real method, including sparse part, AdamW dynamics, changing tensor factors during training and mixed precision setting.
2. Baseline comparison are incomplete:
The experiments compare mainly with Adam, Galore, sparse-only, and low-rank-only methods. Stronger memory-efficient optimizers should also be included to make the empirical claims stronger.
3. Tensor decomposition adds overhead. The paper mentions slowdown, but the tradeoff between memory saving and training time should be discussed more clearly.
4. Novelty of this paper is moderate. The method mainly combines existing ideas: lowrank compression, sparse gradients, Tucker decomposition, and Adam optimization. The comtribution is more an integration for neural operators than a fundamentally new optimization method.

**Requested Changes:**

Major changes:
1. Add comparisons with stronger memory-efficient optimizers or training methods such as Adafactor or other recent efficient optimizers. Current baseline are limited.
2. The current theory analyzes only a simplified low-rank fixed-projection setting. Please clearly state this limitation and discuss how it relates to the full TensorGRaD algorithm.
3. Most experiments focus on neural operators tasks. Please add experiments on other architectures such as CNN, Transformers, to demonstrate its generality.
4. Tensor decomposition adds computational cost. Please add clearer training time comparisons and discuss when the memory/runtime tradeoff is worthwhile.
5. Describe how rank ratios, sparsity ratios, update intervals and scaling coefficients were selected.
6. When TensorGRaD outperforms Adam, please discuss whether this may come from regularization effects due to compression noise.

Minor changes:
1. Add a notation table and more clearly symbols such as compressed gradients, reconstructed gradients, residuals, and final updates.

---

> ### Author Response · Authors · 2026-04-29
>
> We thank the reviewer for the thoughtful feedback. We agree that the previous draft did not make some limits and practical choices sufficiently clear. In the revision, we strengthened the empirical section, clarified the scope of the theory, and tightened the paper’s positioning.
>
> **1. Positioning and novelty (W4/RC4).**
> We agree that TensorGRaD is not a fundamentally new optimizer in the sense of introducing a new Adam-like update rule. Our claim is narrower: TensorGRaD is a tensor-aware optimizer compression method for models with intrinsically higher-order gradients. Its novelty is not just combining low-rank and sparse structure, but (i) extending optimizer compression from matrices to tensor gradients, (ii) showing theoretically and empirically that naive matricization as in GaLore is a poor fit because it destroys mode structure, (iii) introducing a low-rank+sparse residual scheme that improves over pure tensor low-rank compression at matched budgets, and (iv) developing a practical online algorithm with warm-started Tucker updates, sparse-state handling.
>
> **2. Memory-efficient baselines (W2/RC1).**
> We agree that stronger baselines are useful. In the revision, we added SGD and Adafactor on NS128 (Tab. 3). Both perform substantially worse than TensorGRaD, despite using less optimizer memory: in full precision, SGD and Adafactor obtain test L2 losses of 9.09 and 8.19, versus 5.72 for TensorGRaD and 5.66 for Adam; in Mixed-1, they obtain 11.12 and 8.87, versus 5.71 for TensorGRaD and 5.62 for Adam. Details and sweeps are in App B.
>
> **3. Theory scope (W1/RC2).**
> We now make this explicit in the main paper theory section. The analysis covers only the fixed-projection tensor low-rank component of the method; it does not cover the sparse branch, AdamW moments, changing Tucker factors, or mixed precision. We position the theorem as explaining why tensor-mode-preserving projections are structurally preferable to matricization, not as a full convergence theorem for TensorGRaD as implemented.
>
> **4.CNN/Transformer experiments (RC3).**
> We appreciate the suggestion, and have tried to clarify the motivation in our updated manuscript. Our work is specifically motivated by tensor-structured models, with neural operators as the main target: in this setting, gradients and optimizer states are naturally higher-order tensors. This causes several issues specific to this setting: i) optimizer memory is a particularly severe bottleneck at high resolution. ii) naive matricization destroys meaningful mode structure and leads to poor performance, meaning existing approaches cannot be directly applied. By contrast, many standard Transformer parameters are 2D unless one first imposes a tensorization, so the motivation for tensor-mode-preserving compression is less direct. We therefore clarify that the contribution is about tensor-aware optimizer compression for tensor-structured models, and retain TFNO and GINO as evidence that the method is not restricted to one exact FNO setup.
>
> **5. Runtime vs memory (W3/RC4).**
> We moved the update-frequency ablation into the main paper (Fig. 6) and now report both accuracy and runtime as a function of the projector update gap $T$. The revised discussion presents TensorGRaD more clearly as a memory-accuracy-time trade-off. At $T=10^3$, the slowdown is about 13% for pure low-rank compression and 20% for low-rank+sparse compression.
>
> **6. Hyperparameters (RC5).**
> We now make the selection procedure explicit and tie it to the ablations. A simple default recipe is: use a total optimizer budget of about 25% of Adam, assign most of it to the low-rank branch, keep the sparse branch small (typically 2-5%), and choose $T \in [10^2,10^3]$. This is supported by Fig. 3, Fig. 5, and Fig. 6. We also added a short practical hyperparameter guide in App. C. We keep $\lambda$ and $\alpha$ fixed across the main experiments and now state this in the hyperparameter section.
>
> **7. TensorGrad vs Adam (RC6).**
> We agree that this should be discussed more clearly. We had already noted in the previous version that occasional slight improvements over Adam may reflect a regularization effect from compression, and in the revision, we make this interpretation more explicit and tone down stronger language. The new sparse-share ablation is consistent with this view: adding a small sparse residual can help, but pushing sparsity too far hurts.
>
> **8. Notation clarity (RC7).**
> We agree. In the revision, we added a notation table (Tab. 4) defining the main gradient/update tensors and the low-rank/sparse optimizer moments. We also cleaned up notation in the theory section so it is clearer which objects belong to the analyzed low-rank component and which belong to the full method.

---

### Decision · Action_Editor_sAvh · 2026-06-12

**Recommendation:** Accept as is

**Audience:**

Yes

**Audience Explanation:**

Sparse approaches (from pruning to autodiff) are gaining ground in the deep learning world, as GPU memory ranks among the main limiting factors during training. The method outlined here takes existing ideas for weight matrices and generalizes them to tensors, which will be of great interest to ML researchers working on physics and other engineering topics. The presentation is also clear and thorough.

Open-sourcing the experimental code is the one remaining step before the camera-ready version.

**Claims And Evidence:**

Yes

**Claims Explanation:**

The paper suggests using a tensor-native decomposition for neural network weights, extending work done in the 2D case. This idea makes sense and the memory savings are evidenced by a vast array of numerical experiments, including comparisons with the previous GaLore baseline and ablation studies.